# Regulation of photosynthetic electron flow on dark to light transition by ferredoxin:NADP(H) oxidoreductase interactions

Manuela Kramer[1,2], Melvin Rodriguez-Heredia[1], Francesco Saccon[1], Laura Mosebach[3], Manuel Twachtmann[2], Anja Krieger-Liszkay[4], Chris Duffy[1], Robert J Knell[1], Giovanni Finazzi[5], Guy Thomas Hanke[1,2]*

[1]School of Biochemistry and Chemistry, Queen Mary University of London, London, United Kingdom; [2]Department of Plant Physiology, Faculty of Biology and Chemistry, University of Osnabrück, Osnabrück, Germany; [3]Institute of Plant Biology and Biotechnology, University of Münster, Münster, Germany; [4]Institute for Integrative Biology of the Cell (I2BC), CEA, CNRS, Univ Paris-Sud, Université Paris-Saclay, Paris, France; [5]Laboratoire de Physiologie Cellulaire et Végétale, UMR 5168, Centre National de la Recherche Scientifique (CNRS), Commissariat a' l'Energie Atomique et aux Energies Alternatives (CEA), Université Grenoble Alpes, Institut National Recherche Agronomique (INRA), Institut de Recherche en Sciences et Technologies pour le Vivant (iRTSV), CEA Grenoble, Grenoble, France

*For correspondence:
g.hanke@qmul.ac.uk

Competing interests: The authors declare that no competing interests exist.

**Abstract** During photosynthesis, electron transport is necessary for carbon assimilation and must be regulated to minimize free radical damage. There is a longstanding controversy over the role of a critical enzyme in this process (ferredoxin:NADP(H) oxidoreductase, or FNR), and in particular its location within chloroplasts. Here we use immunogold labelling to prove that FNR previously assigned as soluble is in fact membrane associated. We combined this technique with a genetic approach in the model plant Arabidopsis to show that the distribution of this enzyme between different membrane regions depends on its interaction with specific tether proteins. We further demonstrate a correlation between the interaction of FNR with different proteins and the activity of alternative photosynthetic electron transport pathways. This supports a role for FNR location in regulating photosynthetic electron flow during the transition from dark to light.

## Introduction

Photosynthetic carbon assimilation in chloroplasts requires NADPH. This is generated by the enzyme ferredoxin:NADP(H) oxidoreductase (FNR) using electrons from ferredoxin (Fd), which is reduced on excitation of photosystem I (PSI) (*Shin et al., 1963*). The pathway is referred to as 'linear electron flow' (LEF) because the electrons originate from water splitting at photosystem II (PSII) and are transferred in a linear progression to PSI via plastoquinone/plastoquinol ($PQ/PQH_2$) and the cytochrome $b_6f$ complex (Cyt $b_6f$), pumping protons across the thylakoid membrane (*Hill and Bendall, 1960*; *Mitchell, 1975*). Plastocyanin (PC) takes these electrons and re-reduces the oxidised PSI. This proton motive force drives synthesis of ATP, which is also required for carbon fixation among a host of other reactions. FNR shows an extremely high control co-efficient for photosynthetic rate (meaning that changing FNR concentrations strongly influences flux through the pathway) in tobacco (*Nicotiana tabacum*): 0.7 in limiting light and 0.94 in saturating light (*Hajirezaei et al., 2002*).

Chloroplasts are bounded by an envelope membrane and contain a soluble matrix (the stroma) and an internal membrane (the thylakoid) composed of two distinct domains: stacks of appressed discs known as grana, connected by sheets of membrane called lamellae (*Paolillo, 1970*; *Mustárdy et al., 2008*). Apart from the electron shuttling proteins PC and Fd, all components of LEF are unambiguously localised to the thylakoid membrane except FNR, whose location remains controversial.

Although it has been found associated with PSI (*Andersen et al., 1992*), it is suggested that membrane association is unnecessary for FNR function (*Benz et al., 2010*). This is because mutation of two, higher plant specific, FNR tethering proteins (Tic62 and TROL) caused all FNR to be soluble but had little impact on LEF (*Lintala et al., 2014*). Current dogma therefore states that $NADP^+$ photoreduction is conducted by soluble FNR, free in the stroma (*Benz et al., 2010*), and indeed a large proportion of FNR is recovered as a soluble protein after cell fractionation of algae (*Mosebach et al., 2017*) and higher plants (*Hanke et al., 2005*; *Okutani et al., 2005*; *Böhme, 1978*). Contrary to this theory, FNR enzyme activity increases when the protein is associated with the membrane (*Carrillo and Vallejos, 1982*; *Forti and Bracale, 1984*).

PSI can be re-reduced by returning electrons from Fd to PQ, thus pumping protons without generating NADPH. This cyclic electron flow (CEF) protects the photosynthetic machinery when there is an imbalance between the production of excited electrons and their consumption (*Huang et al., 2018*; *Yamori et al., 2016*), and regulates the ratio between ATP and NADPH produced (*Suorsa et al., 2012*; *Walker et al., 2014*; *Munekage et al., 2010*; *Joliot and Johnson, 2011*; *Kramer and Evans, 2011*). Two CEF pathways have been defined: one is catalysed by a homologue of respiratory complex I called NDH (*Burrows et al., 1998*; *Shikanai et al., 1998*; *Kofer et al., 1998*); the other is defined by its sensitivity to antimycin A, and depends on the Pgr5 (*Munekage et al., 2002*) and PgrL1 proteins (*DalCorso et al., 2008*). There are some differences in CEF pathways between angiosperms, which retain the NDH complex, and algae such as *Chlamydomonas reinhardtii* which replace it with an NDH2 type, monomeric complex (*Peltier et al., 2016*). Curiously, FNR is implicated in the antimycin A sensitive pathway, being found in complex with PgrL1 in the green alga *Chlamydomonas* (*Iwai et al., 2010*) and identified as a PgrL1 interaction partner in higher plants (*DalCorso et al., 2008*). Moreover, *pgr5* knockout in *Chlamydomonas* increases FNR solubilisation (*Mosebach et al., 2017*). There is also some evidence supporting a function for FNR in higher plant CEF from inhibitor studies (*Ravenel et al., 1994*; *Shahak et al., 1981*; *Bendall and Manasse, 1995*; *Mills et al., 1979*; *Cleland and Bendall, 1992*; *Ye and Wang, 1997*; *Hosler and Yocum, 1985*), but a definitive mutant study is hampered by the severe phenotype when $NADP^+$ reduction is disrupted (*Bendall and Manasse, 1995*; *Lintala et al., 2012*).

Higher plant chloroplasts switch rapidly between LEF and CEF in response to environmental conditions (*Joliot and Joliot, 2005*; *Asada et al., 1993*) and dynamic transfer of FNR between membrane complexes has been proposed as part of this mechanism (*Joliot and Johnson, 2011*; *Breyton et al., 2006*). FNR iso-proteins with variable capacity for membrane association could help to test this hypothesis and have been identified so far in wheat, Arabidopsis, and maize (*Hanke et al., 2005*; *Okutani et al., 2005*; *Gummadova et al., 2007*). Arabidopsis has two FNR iso-proteins (AtFNR1 and AtFNR2), and knock-out of the At*FNR1* gene results in complete solubilisation of AtFNR2, indicating cooperativity in membrane association of the two FNR iso-proteins (*Hanke et al., 2008*; *Lintala et al., 2007*). Maize has 3 FNR iso-proteins, which are differentially localised between cell types engaging predominantly in CEF or LEF (*Twachtmann et al., 2012*). Remarkably, when we expressed maize FNRs heterologously in Arabidopsis, they showed specific association with either TROL (ZmFNR1), Tic62 (ZmFNR2), or were soluble (ZmFNR3). For clarity, the properties of the different Arabidopsis and maize FNRs are summarised in *Supplementary file 1*.

The grana and lamellae domains of the thylakoid vary in protein composition (*Andersson and Anderson, 1980*), with PSII and its antennae in the appressed grana, and PSI, NDH, and the ATPase restricted to stroma facing regions (*Daum et al., 2010*). Cyt $b_6 f$ is evenly distributed (*Allred and Staehelin, 1986*). As LEF requires PSI and PSII, it is thought to predominantly occur at border regions, where appressed and stroma facing membranes coincide, bringing the necessary components into proximity (*Anderson et al., 2012*). By contrast, the CEF components PSI, PgrL1/Pgr5 (*Hertle et al., 2013*), NDH (*Lennon et al., 2003*), and Cyt $b_6 f$ are all present on the lamellae (*Anderson, 1992*; *Chow et al., 2005*). In order to understand whether FNR can act as part of a regulatory switch between LEF and CEF, it is therefore critical to understand (1) where it is located within the

chloroplast, (2) how this relates to its interaction with different tether proteins, and (3) how these factors impact on the different electron transport pathways. Fluorescence microscopy with FNR is hampered by difficulties in labelling the protein – the N-terminal is critical for interaction with the membrane (*Twachtmann et al., 2012*), while the carboxy-group of the C-terminal Tyr is part of the catalytic mechanism (*Tejero et al., 2005*). In this study we therefore undertook a rigorous immunogold-label (IGL) study on FNR location in Arabidopsis chloroplasts. The data indicate that the enzyme is almost exclusively membrane bound, even in genotypes where it was previously thought to be totally soluble. Moreover, to dissect the connection between interaction, location, and function, we have introduced genes for the different ZmFNR proteins into the *fnr1* mutants, creating Arabidopsis plants with approximately wild-type levels of FNR but enriched in specific interactions: either Tic62-bound, TROL-bound, or soluble. The data show that FNR:protein tether interactions change FNR sub-chloroplast distribution and impact on the dominant CEF pathway that occurs during the transition from dark to light.

## Results

In order to establish a rigorous protocol for FNR localisation by IGL, we first confirmed our antibodies are highly specific (*Figure 1—figure supplement 1*). Then we performed transmission electron microscopy (TEM) IGL on multiple chloroplasts from one individual. We defined sub-compartments in the chloroplast and calculated the FNR staining density within them as particles per $\mu m^2$ (*Figure 1—figure supplement 2*). The minimum number of chloroplasts necessary for a statistically sound interpretation was identified using a power analysis of this data in a mixed model test (chosen as a statistical test because the FNR density in different sub-compartments of each chloroplast is related). This was defined as three chloroplasts per individual, but we have analysed 15 to generate additional statistical power (*Supplementary file 2a*). The distance between a label and its target is influenced by the size of antibodies – meaning a label can potentially be up to 30 nm away from a protein (*Hermann et al., 1996*). To ensure a conservative estimate of membrane localisation, we therefore defined an area approximately 10 nm either side of the membrane as 'membrane bound', and divided chloroplasts into three sub-compartments: stroma, grana core, and combined stromal exposed membranes (lamellae + margins). FNR staining density is five times higher at the lamellae/margin region than in the stroma or the grana core (*Figure 1—figure supplement 2*), which the mixed model describes as highly significant (*Supplementary file 2a*). Although FNR-staining in both grana core and stroma is higher than in the cytosol, we consider this density likely originates from FNR at the lamellae/margin region, labelled by antibody oriented such that it extends beyond the defined 10 nm area either side of the TEM visualised membrane. This interpretation is based on the following reasons: (1) FNR cannot enter the grana core, due to spatial restriction, and so grana assignments must originate from antibodies tethered to FNR at the grana margins. (2) There is no statistical difference between staining density in the grana and stroma, meaning that even the low level detected in the stroma is also likely due to the orientation of label attached to membrane associated FNR. (3) Cyt *f* is also detected at significantly higher density in the stroma than the cytosol by IGL-TEM (*Figure 1—figure supplement 2*, *Supplementary file 2b*). Cyt *f* is part of the Cyt $b_6f$, an integral membrane protein that is never found in the stroma. Nevertheless, we retained the original 10 nm area to define membrane association, as this appears to detect the great majority of membrane bound proteins. We consider that a small amount of false negative assignments to grana core and stroma are preferable to the risk of false positives associated with extending the membrane assignment zone from 10 to 30 nm. As expected, there is no significant difference in Cyt *f* staining between different thylakoid domains. Based on this analysis, we conclude that chloroplasts contain very little soluble, stromal FNR.

It has been reported that during some fixation procedures for immunolabelling whole mammalian cells, soluble cytosolic proteins may be disproportionately washed out of samples in comparison to membrane bound proteins (*Huebinger et al., 2018*). To confirm that this is not the reason for the absence of soluble-localised FNR in our sections, we therefore repeated the experiment to compare several individuals of Wt and *fnr1*, a genotype in which the remaining FNR (AtFNR2) is 100% soluble following mechanical disruption to extract proteins from leaves (*Hanke et al., 2008*; *Lintala et al., 2007*). Again, our analysis shows significantly higher label density of the margins/lamellae than the stroma in both genotypes (*Figure 1*, *Table 1*). There is no significant difference between labelling

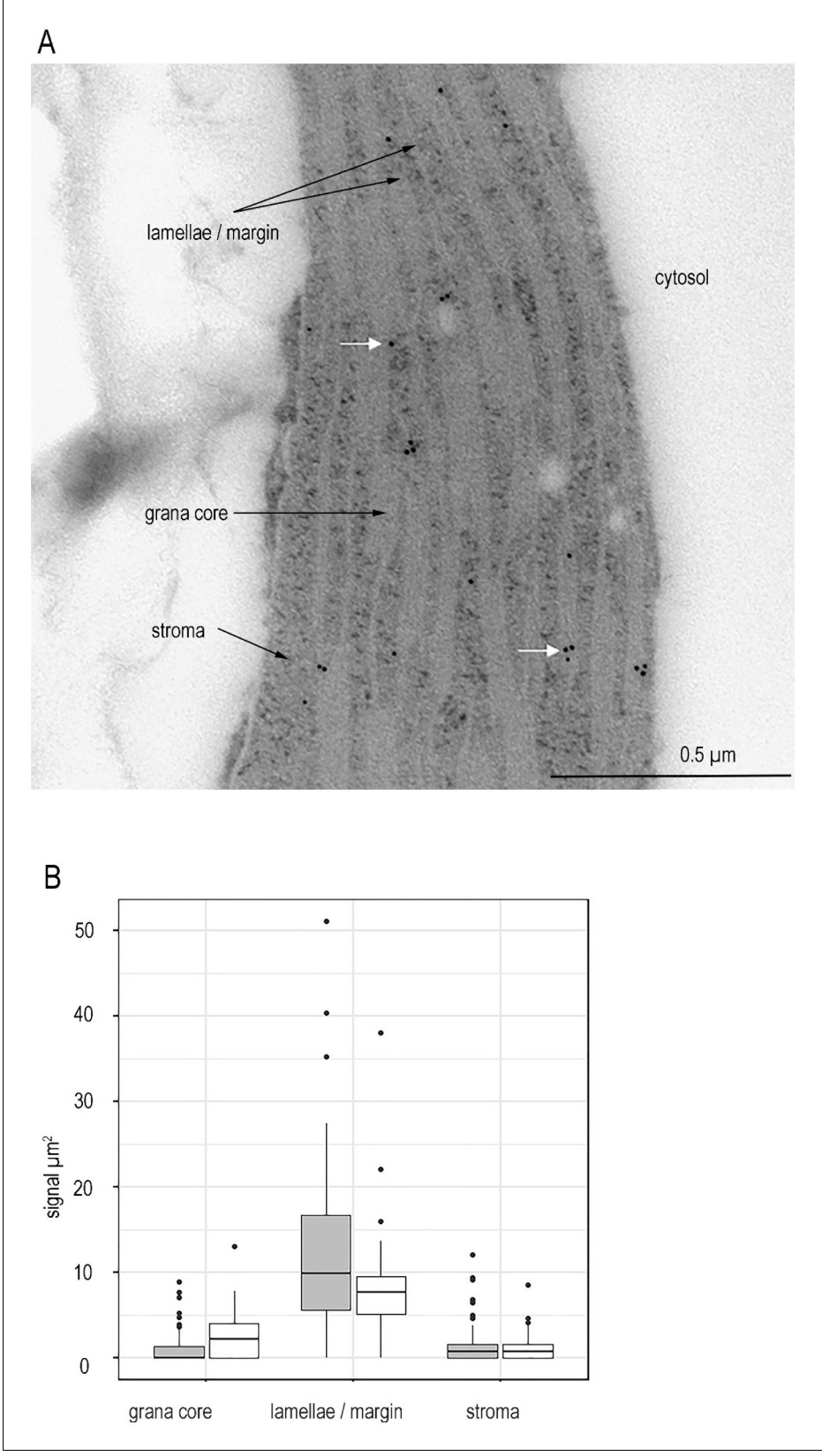

**Figure 1.** FNR is not detected in the chloroplast stroma of higher plants. (**A**) Representative micrograph showing immunogold labelling of FNR in sections of chloroplasts from Wt Arabidopsis detected by IGL-TEM. White arrows indicate example gold particles. (**B**) Immunogold labelling density of FNR in different sub-chloroplast

*Figure 1 continued on next page*

*Figure 1 continued*

compartments, n = 15–22 chloroplasts each from three Wt individuals (grey) and 3 *fnr1* (white) individuals. Outliers shown. See *Figure 1—figure supplement 1* for quality control of antibodies and *Figure 1—figure supplement 2* for optimisation of IG-TEM technique. See *Table 1* for statistical significance in a mixed effects model analysis of variance between genotypes and between sub-compartments within each genotype.

The online version of this article includes the following figure supplement(s) for figure 1:

**Figure supplement 1.** Specificity of antisera used for Immunogold labelling and blue native PAGE western blots.
**Figure supplement 2.** Detecting protein localisation in sub-chloroplast compartments.

density of stroma and grana core, indicating that there is negligible FNR in the stroma. This result, combined with previous observations that organellar proteins are much less prone to loss during fixation than cytosolic proteins (*Schnell et al., 2012*), strongly suggests that the FNR distribution we have measured reflects the situation in the native chloroplast. The studies that previously determined completely soluble distribution of FNR in the *fnr1* mutant were based on mechanical separation followed by centrifugation. By contrast, our IGL data indicate that in situ almost all FNR is thylakoid associated. The FNR previously assigned as soluble must therefore be associated with the membrane through weak associations that are disrupted during mechanical extraction.

To try to understand more about the location of tightly bound and weakly associated FNR at the thylakoid, we analysed plants where FNR is localised to different membrane complexes. We exploited genes for three maize FNR proteins (ZmFNR1, ZmFNR2, and ZmFNR3) with variable affinity to the TROL and Tic62 membrane tethers (*Twachtmann et al., 2012*). These were expressed in the Arabidopsis *fnr1* mutant, under control of the Arabidopsis *FNR1* promoter – resulting in approximately wild-type FNR protein contents. Western blots to show FNR proteins in these lines are presented in *Figure 2*. SDS-gels (*Figure 2A*) separate proteins according to mass, while native gels (*Figure 2B*) separate proteins according to native charge and retain some strong protein:protein interactions. As previously reported, mechanical separation of supernatant and pellet fractions from *fnr1* leaves caused recovery of all FNR in the soluble fraction. We have now termed this 'weakly bound' FNR, based on the results in *Figure 1*. In the *fnr1*-ZmFNR1 genotype, ZmFNR is recovered in both the weakly bound and the pellet (now termed 'tightly bound') fractions. The tightly bound ZmFNR1 is mostly in high molecular weight complexes (*Figure 2B*) associated with TROL (*Figure 2C*). Interestingly, when we confirmed the specificity of our TROL antibody, we noted that total TROL abundance partly correlates with the intensity of the BNP band that reacts with both FNR and TROL antisera (*Figure 2C*), being increased in the *fnr1*-ZmFNR1 line and decreased in the *fnr1*, *fnr1*-ZmFNR2, and *fnr1*-ZmFNR3 lines (*Figure 1—figure supplement 1*). Heterologously expressed ZmFNR2 is also found in both weakly associated and strongly bound fractions, and its expression results in the rescue of native AtFNR2 recruitment to the tightly bound membrane fraction in the *fnr1* genotype (*Figure 2A* lower band). This lends support to the hypothesis that co-operative interactions with other FNR iso-proteins are necessary to recruit AtFNR2 to Tic62 and TROL tethers (*Hanke et al., 2008*; *Lintala et al., 2009*; *Lintala et al., 2007*). FNR is enriched at Tic62 in this line (*Figure 2C*). In the *fnr1*-ZmFNR3 line, nearly all FNR remains weakly associated (*Figure 2A*).

The plants expressing genes for the maize FNR1, FNR2, and FNR3 iso-proteins in an Arabidopsis *fnr1* mutant background (*fnr1*-ZmFNR1, *fnr1*-ZmFNR2, and *fnr1*-ZmFNR3) were then analysed by IGL-TEM (*Figure 3*). In this case we further divided the staining density of lamellae/margin into separate margin (any stromal facing membrane adjacent to an appressed membrane) and lamellae (all thylakoid membrane not adjacent to appressed membrane) areas. See *Figure 1—figure supplement 2* for an example of domain area labelling. We consider this a primarily functional, rather than structural, distinction, as many regions we have classed as margin will not be curved, but should have PSI and PSII in relatively close proximity. We examined the possibility that differences in chloroplast ultrastructure between genotypes might influence our findings, by comparing the relative size of our defined areas between the genotypes (*Figure 3—figure supplement 1*). We found that the *fnr1* mutant shows a small increase in relative stromal area and corresponding decrease in relative margin area. The relative areas of chloroplast sub-compartments did not differ from the Wt in any of the other genotypes. Analysis of FNR staining density in chloroplasts of all genotypes is shown in *Figure 3* (statistics in *Table 2*). Absolute staining density might be influenced by variation in FNR-

**Table 1.** Mixed effects model investigating changes in FNR density between total chloroplast sub-compartments of three individuals each of WT and *fnr1* Arabidopsis.

Analysis of data presented in *Figure 1*. Fixed effects taking either label density in the stroma as the intercept or label density in the margins/lamellae as the intercept. Linear mixed model fit by REML. Signif. codes: 0 '***' 0.001 '**' 0.01 '*' 0.05 '.' 0.1 ' ' 1.

**Deletion test** carried out using Satterthwaite's method with the R package lmerTest (Kuznetsova, Brockhoff & Christensen 2017). The model is a mixed effects model with random intercepts. The square root of response is the response variable, tissue is the fixed effect, and individual the random effect.

| Fixed effect deleted | Sum sq | Mean sq | Num DF | Den DF | F value | Pr (>F) | |
|---|---|---|---|---|---|---|---|
| Sub-compartment | 14.231 | 7.1153 | 2 | 295.58 | 7.4565 | 0.000693 | *** |

Model summary:

Random effects:

| Groups | Name | Variance | Std. Dev. |
|---|---|---|---|
| individual | (Intercept) | 0.1896 | 0.4354 |
| Residual | | 0.9542 | 0.9769 |

Number of obs: 306, groups: individual, 6

Fixed effects when *fnr1* stroma is set as the intercept

| | Estimate | Std. Error | DF | t value | Pr (>|t|) | |
|---|---|---|---|---|---|---|
| (Intercept) | 0.9165 | 0.2791 | 4.2686 | 3.283 | 0.0276 | * |
| Grana | −0.3607 | 0.1714 | 295.5794 | −2.105 | 0.0361 | * |
| Margin/lamellae | 2.2884 | 0.1714 | 295.5794 | 13.355 | <$2^{-16}$ | *** |
| Genotype comparison WT:*fnr1* | 0.1230 | 0.4128 | 5.0375 | 0.298 | 0.7776 | |
| WT grana: *fnr1* grana | 0.6286 | 0.2845 | 295.5794 | 2.210 | 0.0279 | * |
| WT margin/lamellae: *fnr1* margin/lamellae | −0.4661 | 0.2845 | 295.5794 | −1.638 | 0.1025 | |

Fixed effects when *fnr1* margin/lamellae is set as the intercept

| | Estimate | Std. Error | DF | t value | Pr (>|t|) | |
|---|---|---|---|---|---|---|
| (Intercept) | 0.9165 | 0.2791 | 4.2686 | 11.482 | 0.000228 | *** |
| Grana | −2.6491 | 0.1714 | 295.5794 | −15.460 | <$2^{-16}$ | *** |
| Stroma | −2.2884 | 0.1714 | 295.5794 | −13.355 | <$2^{-16}$ | *** |
| Genotype comparison WT:*fnr1* | −0.3430 | 0.4128 | 5.0375 | −0.831 | 0.443524 | |
| WT grana: *fnr1* grana | 1.0947 | 0.2845 | 295.5794 | 3.848 | 0.000146 | *** |
| WT stroma: *fnr1* stroma | −0.4661 | 0.2845 | 295.5794 | 1.638 | 0.102459 | |

Fixed effects when Wt stroma is set as the intercept

| | Estimate | Std. Error | DF | t value | Pr (>|t|) | |
|---|---|---|---|---|---|---|
| (Intercept) | 1.0395 | 0.3041 | 5.8576 | 3.418 | 0.0147 | * |
| Grana | 0.2679 | 0.2271 | 295.5794 | 1.180 | 0.2391 | |
| Margin/lamellae | 1.8223 | 0.2271 | 295.5794 | 8.024 | $2.43^{-14}$ | *** |
| Genotype comparison WT:*fnr1* | −0.1230 | 0.4128 | 5.0375 | −0.298 | 0.7776 | |
| Wt grana: *fnr1* grana | −0.6286 | 0.2845 | 295.5794 | −2.210 | 0.0279 | * |
| Wt margin/lamellae: *fnr1* margin/lamellae | 0.4661 | 0.2845 | 295.5794 | 1.638 | 0.1025 | |

Fixed effects when Wt margin/lamellae is set as the intercept

| | Estimate | Std. Error | DF | t value | Pr (>|t|) | |
|---|---|---|---|---|---|---|
| (Intercept) | 2.8618 | 0.3041 | 5.8576 | 9.411 | $9.41^{-05}$ | *** |
| Grana | −1.5544 | 0.2271 | 295.5794 | −6.844 | $4.44^{-11}$ | *** |
| Stroma | −1.8223 | 0.2271 | 295.5794 | −8.024 | $2.43^{-14}$ | *** |

*Table 1 continued on next page*

Table 1 continued

**Deletion test** carried out using Satterthwaite's method with the R package lmerTest (Kuznetsova, Brockhoff & Christensen 2017). The model is a mixed effects model with random intercepts. The square root of response is the response variable, tissue is the fixed effect, and individual the random effect.

| Fixed effect deleted | Sum sq | Mean sq | | Num DF | Den DF | F value | Pr (>F) | |
|---|---|---|---|---|---|---|---|---|
| Genotype comparison WT:*fnr1* | 0.343 | 0.4128 | | 5.0375 | | 0.831 | 0.443524 | |
| Wt grana: *fnr1* grana | −1.0947 | 0.2845 | | 295.5794 | | −3.848 | 0.000146 | *** |
| Wt stroma: *fnr1* stroma | −0.4661 | 0.2845 | | 295.5794 | | −1.638 | 0.102459 | |

isoform antigenicity, so we have compared only between sub-chloroplast domains of each genotype. The greatest variation in distribution between genotypes is seen in the density of FNR staining associated with the lamellae (blue boxes in *Figure 3*). While FNR density in the lamellae region is similar to that at the margin for Wt chloroplasts, the *fnr1* mutant shows a significant decrease in label density at the lamellae relative to the margins (*Table 2*). Introduction of ZmFNR1 (TROL binding) to the *fnr1* background restored equal FNR density between the margins and lamellae regions. By contrast, introduction of ZmFNR2 (Tic62 binding) to the *fnr1* background results in much higher label density at the stromal lamellae relative to the margins. This tendency is also seen on introduction of ZmFNR3 (weak binding), but with lower statistical significance. We interpret the dramatic change in FNR localisation on ZmFNR2 expression in the *fnr1* background as being related to the restoration of native AtFNR2 recruitment into strong interactions at the thylakoid membrane in this genotype (*Figure 2A*).

It has been suggested that the location of FNR at different thylakoid complexes might regulate electron transport (*Breyton et al., 2006*; *Joliot and Johnson, 2011*; *Twachtmann et al., 2012*), and we therefore measured whether sub-chloroplast FNR distribution had an impact on enzyme activity in $NADP^+$ reduction. $NADP^+$ reduction and NADPH oxidation kinetics were followed in chloroplasts isolated from the different genotypes over a short illumination period followed by a dark period (*Figure 4*). Although amplitudes of fluorescence are quite consistent between genotypes (*Figure 4*), comparisons of absolute NADPH concentrations between chloroplast preparations are problematic, as the proportion of broken chloroplasts may vary. However, these data do allow us to accurately compare the kinetics of reduction and oxidation.

As described previously for pea chloroplasts (*Schreiber and Klughammer, 2009*; *Latouche et al., 2000*) and Arabidopsis (*Hanke et al., 2008*), isolated chloroplasts show distinct components of $NADP^+$ reduction. In the measurements shown in *Figure 4*, only two components are clearly observed: a fast one of less than 1 s, and a slow one that lasts ~20 s. Further characterisation of these components is described in Appendix 1. The two observed components of $NADP^+$ reduction can be fitted (*Supplementary file 2c*) and vary significantly in relative size between genotypes (*Supplementary file 2d*), depending on the abundance and location of FNR. The *fnr1* mutant and the genotype expressing weakly bound maize FNR in the mutant background (*fnr1*-ZmFNR3) show a relatively small contribution of the slow component when compared to Wt, *fnr1*-ZmFNR1, and *fnr1*-ZmFNR3. NADPH oxidation in the dark was well fit to a single component using a Hill coefficient and no difference between genotypes was detected (*Supplementary file 2e*).

We then determined that these differences in NADP(H) kinetics are not due to pleiotropic effects altering the abundance of other proteins that could impact photosynthetic electron transport or NADP(H) poise (*Figure 4—figure supplement 1*). We found no differences in protein abundance that correlate with those in NADP(H) reduction and oxidation kinetics. Although this does not discount the possibility that variable FNR:FNR tether interactions could influence regulation of these proteins (see later discussion), it does allow us to discount pleiotropic changes in their total capacity.

It is reported that FNR interactions with Tic62 and TROL are weakened by exposure to light (*Alte et al., 2010*; *Benz et al., 2009*). This means that during the illumination period of our NADP (H) measurement, these interactions will be weakened. Because interaction with Tic62 and TROL is not detected in the *fnr1* and *fnr1*-ZmFNR3 genotypes (*Figure 2C*), which also have a diminished slow phase of $NADP^+$ reduction (*Figure 4*), we propose that the release of FNR from tightly bound tether locations could contribute to an increase in the activity of the enzyme. The kinetics in *Figure 4* suggest that either (1) strongly bound FNR (prevalent after dark adaptation) has less efficient

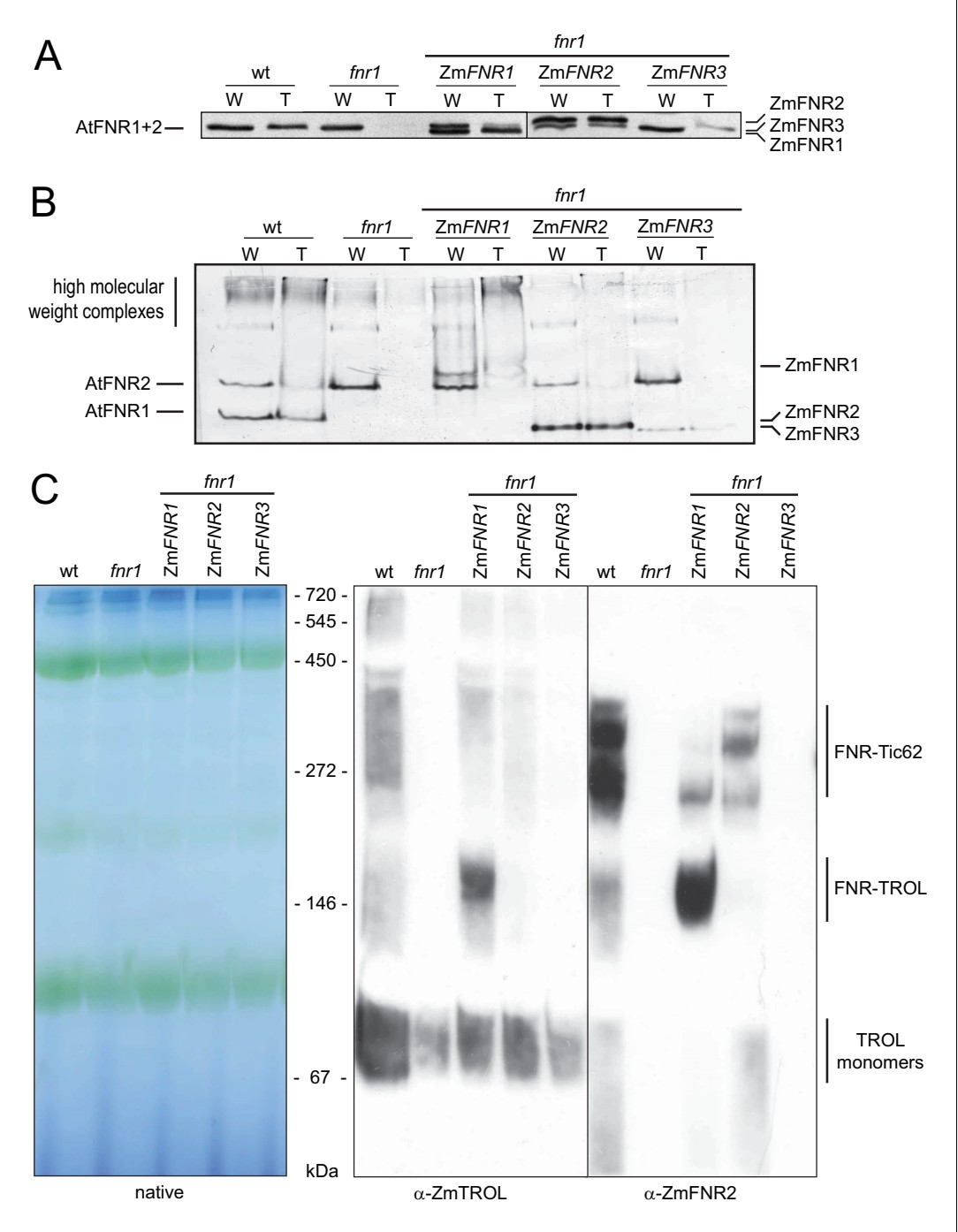

**Figure 2.** Arabidopsis plants with variable FNR:membrane tether interactions. Leaf extracts of Arabidopsis wt, *fnr1*, and *fnr1* mutants transformed to express genes for the maize FNR proteins ZmFNR1, ZmFNR2, and ZmFNR3 in the *fnr1* background were separated into soluble and insoluble fractions. These were designated loose (L) and tight (T) membrane bound FNR fractions, based on the analysis in *Figure 1*. Samples were subjected to (**A**) SDS-PAGE (25 µg protein prior to separation of L and T fractions) or (**B**) native PAGE (20 µg protein prior to separation of L and T fractions) before immunoblotting, challenge with antisera against FNR, then detection with alkaline phosphatase. Migration positions of Arabidopsis (At) FNRs and maize (Zm) FNRs are indicated to the left and right, respectively. (**C**) Recruitment of maize FNR proteins into specific Arabidopsis thylakoid membrane complexes. Chloroplasts were isolated from the same genotypes used in (**A**) and (**B**). Thylakoid membranes were solubilised with DDM and subjected to blue native-PAGE (BNP). Samples were loaded on an equal chlorophyll basis, with 3.2 µg per lane, western blotted and challenged with antisera raised against FNR (rabbit) and TROL

*Figure 2 continued on next page*

*Figure 2 continued*

(guinea pig) before visualisation using secondary antisera conjugated to horseradish peroxidase (chemiluminescence) and alkaline phosphatase respectively. Positions of molecular mass markers are indicated between the gels and the blots in kDa.

NADP$^+$ reduction activity, and that dissociation of FNR from Tic62 and TROL therefore represents a mechanism for upregulation, or (2) there is a difference between the genotypes in the speed with which they upregulate downstream NADPH consumption processes.

We then assessed the impact of FNR location on photosynthetic electron transport. Because of the longstanding debate about the role of FNR in CEF (*Bendall and Manasse, 1995*; *Bojko et al., 2003*; *Buchert et al., 2018*; *Hanke et al., 2008*; *Hertle et al., 2013*; *Hosler and Yocum, 1985*; *Iwai et al., 2010*; *Mosebach et al., 2017*; *Shahak et al., 1981*; *Ye and Wang, 1997*; *Zhang et al., 2001*), and the suggestion that relocation of FNR might be a mechanism to switch between LEF and CEF (*Breyton et al., 2006*; *Joliot and Johnson, 2011*), we paid particular attention to CEF related parameters. It is reported that rates of CEF are highest in the first 20 s of illumination following dark adaptation (*Joliot and Joliot, 2005*), and this corresponds to the time scale over which the slow kinetic phase of NADP$^+$ reduction kinetics develops (*Figure 4*). We therefore compared the electro-chromic band shift (ECS), here used to quantify electron flow rates, following either 20 s or 5 min acclimation to actinic light (*Figure 5A*). To differentiate LEF from CEF, we applied either a pulse of actinic light illumination (stimulation of both PSI and PSII, and therefore CEF + LEF), or a pulse of far red light illumination (PSI excitation only, and therefore CEF only).

As an alternative method, we also performed the experiment following infiltration with DCMU during dark adaptation. This is a specific PSII inhibitor, which showed the same trend, but with much greater variation (*Figure 5—figure supplement 1*). When the data are normalised to a single turn-over flash, it is possible to calculate rates of CEF/total EF (*Figure 5B*, see Materials and methods for a detailed explanation on the calculation procedure). As expected, for WT, there is significantly

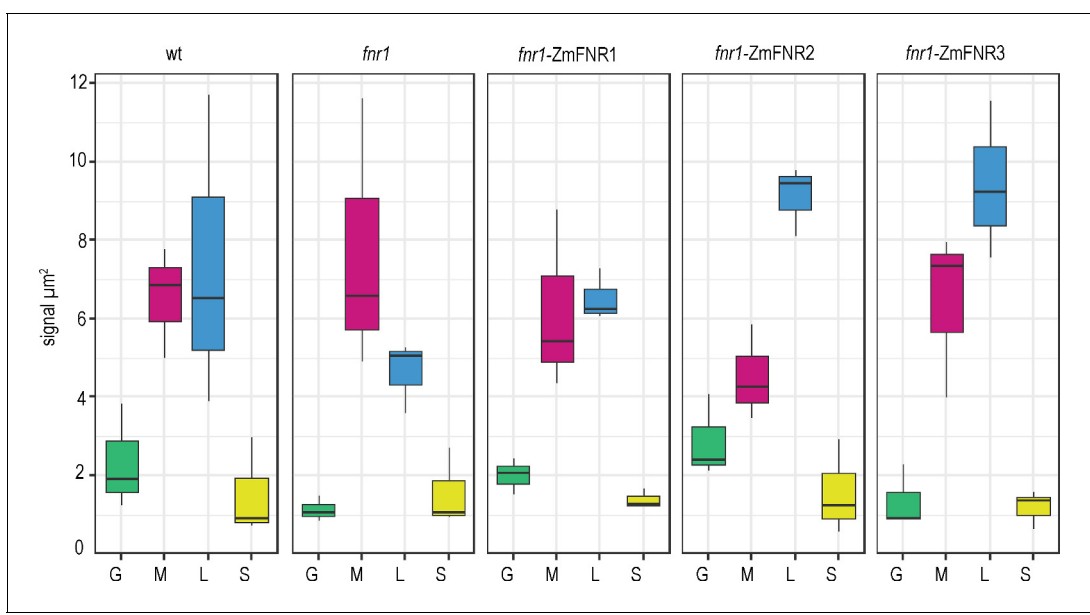

**Figure 3.** Tether interactions determine FNR sub-chloroplast location. Density of immunogold labelled FNR in different sub-chloroplast compartments of the indicated genotypes. Values are averages of three biological replicates, with combined label and area for the sub-compartments of 15–20 chloroplasts for each individual. Areas shown are: grana core (green), margins (magenta), stromal lamellae (blue), and stroma (yellow). See *Figure 3— figure supplement 1* for confirmation that sub-chloroplast areas do not vary between genotypes. Statistical significance between sub-compartment signal density within genotypes was calculated with a mixed effects model analysis of variance (*Table 2*).
The online version of this article includes the following figure supplement(s) for figure 3:

**Figure supplement 1.** Comparison of areas of different chloroplast sub-compartments between genotypes.

more CEF following a short (20 s) light exposure in dark adapted leaves than after light acclimation. This trend is not seen in the *fnr1* mutant, suggesting it lacks the capacity to upregulate CEF in dark adapted leaves. Heterologous expression of either tether bound FNR, ZmFNR1, or ZmFNR2 rescues this capacity, but the more weakly interacting ZmFNR3 does not. All genotypes show similar CEF/total EF values after light acclimation, indicating that the impact of FNR location is related to the tether interactions that occur following dark adaptation. Western blotting confirms that there is little difference in abundance of major PET components between the genotypes (*Figure 4—figure supplement 1*). In several replicated experiments the only consistent differences seen are decreased PsbA in the *fnr1* line and increased PC in the *fnr1*-ZmFNR3 line. Critically, there is no difference in abundance of PgrL1 or subunit 1 of the NDH complex, indicating that CEF components are equivalent and the variation seen in our measurements of CEF is related to FNR location rather than secondary effects.

We also compared activity of PSI (P700 oxidation) and PSII (chlorophyll fluorescence) in the same genotypes under identical conditions to the ECS experiment, in order to understand how these parameters relate to the ECS. *Figure 5C* shows selected parameters, measured after a 20 s high light treatment of either dark adapted or light acclimated plants. PSI activity (ΦI) during high light treatment is higher after light acclimation and similar between all genotypes except for a small decrease in *fnr1*-ZmFNR3 in dark adapted plants. As expected under high light treatment, acceptor limitation at PSI (Y(NA)) is high, and this is ameliorated somewhat by light adaptation. Unexpectedly, after dark adaptation the *fnr1* mutant, which theoretically has lower Fd oxidation capacity, shows lower acceptor limitation than the other genotypes. Correspondingly, measurements of (Y(ND)) show significantly higher limitation in donors to PSI for the *fnr1* mutant than all other genotypes.

These data are consistent with the deficiency in CEF seen in dark adapted *fnr1* (*Figure 5A*), with fewer electrons being cycled back to the donor side of PSI. The opposite trend is seen following light acclimation, with *fnr1* showing higher acceptor limitation and lower donor limitation. This is consistent with decreased Fd oxidation capacity at PSI resulting in decreasing acceptor availability. The *fnr1*-ZmFNR3 plants, which also appear deficient in CEF (*Figure 5A*), do not show a corresponding donor side limitation at PSI following dark adaptation. This discrepancy could be related to the elevated levels of PC protein in the *fnr1*-ZmFNR3 plants (*Figure 4—figure supplement 1*), as PC is the electron donor to PSI. As expected, light incubation leads to much greater LEF flux, with an increase in PSII acceptor availability as seen in ΦII and qL for all genotypes. Taken together, our data support a model where FNR location regulates electron transport through both LET and CET pathways.

To test this further, we repeated our IGL-TEM experiment to examine sub-chloroplast FNR localisation following light adaptation. These data are plotted in *Figure 6* in a comparison with those generated in the original dark adapted experiment from *Figure 3*. Unlike the dark adapted leaves, where staining density is equivalent between margins and lamellae, light adaptation gives higher FNR density at the margins than in the lamella. FNR density is significantly higher in the margins of light acclimated than dark adapted plants (*Table 3*), a region proposed to be highly active in LEF (*Anderson, 1992*; *Chow et al., 2005*).

## Discussion

### FNR is not free in the stroma as a soluble protein

Until now it has been the general consensus that in higher plant chloroplasts a significant proportion of FNR is soluble (*Carrillo and Vallejos, 1982*; *Lintala et al., 2007*; *Okutani et al., 2005*; *Shin et al., 1963*; *Matthijs et al., 1986*; *Gummadova et al., 2007*). Here we present strong evidence that this is not the case (*Figure 1—figure supplement 2*, *Figure 1*, *Figure 3*, *Figure 6*, *Tables 1–3*, *Supplementary files 2a, 2b*) with only membrane bound FNR detected, even in genotypes where FNR was previously considered totally soluble (*Hanke et al., 2008*; *Lintala et al., 2007*). The most likely explanation for this is that the aggressive cell disruption procedures, or osmotic shock followed by solubilisation for BNP, disrupt weak associations of FNR with other membrane complexes (*Andersen et al., 1992*; *Zhang et al., 2001*), or possibly even the membrane itself (*Grzyb et al., 2018*; *Grzyb et al., 2008*). We have therefore redefined soluble FNR as weakly associated FNR. These results are in good agreement with an IGL study showing FNR is only found

**Table 2.** Mixed effects model investigating changes in FNR density between total chloroplast sub-compartments of three individuals each from Arabidopsis genotypes WT, *fnr1* mutant and expressing Zm*FNR1*; Zm*FNR2* and Zm*FNR3* in the *fnr1* background.

Analysis of data presented in **Figure 3**. Fixed effects taking either label density in the stroma as the intercept or label density in the margins/lamellae as the intercept. Linear mixed model fit by REML. Signif. codes: 0 '***' 0.001 '**' 0.01 '*' 0.05 '.' 0.1 ' ' 1.

Wt
Deletion test carried out using Satterthwaite's method with the R package lmerTest (Kuznetsova, Brockhoff & Christensen 2017).
The model is a mixed effects model with random intercepts. The square root of response is the response variable, tissue is the fixed effect, and individual the random effect.

| Fixed effect deleted | Sum Sq | Mean Sq | Num DF | Den DF | F value | Pr (>F) | |
|---|---|---|---|---|---|---|---|
| Sub-compartment | 4.991 | 1.6637 | 3 | 6 | 36.152 | 0.0003089 | *** |

Model summary:

Random effects:

| Groups | Name | Variance | Std. Dev. | | | | |
|---|---|---|---|---|---|---|---|
| Individual | (Intercept) | 0.21097 | 0.4593 | | | | |
| Residual | | 0.04602 | 0.2145 | | | | |

Number of obs: 12, groups: individual, 3

Fixed effects when stroma is set as the intercept:

| | Estimate | Std. Error | DF | t value | Pr (>|t|) | |
|---|---|---|---|---|---|---|
| (Intercept) | 1.1743 | 0.2927 | 2.6475 | 4.012 | 0.034961 | * |
| Grana | 0.3104 | 0.1752 | 6 | 1.772 | 0.126711 | |
| Lamellae | 1.4748 | 0.1752 | 6 | 8.42 | 0.000153 | *** |
| Margin | 1.3736 | 0.1752 | 6 | 7.842 | 0.000227 | *** |

Fixed effects when lamellae is set as the intercept:

| | Estimate | Std. Error | DF | t value | Pr (>|t|) | |
|---|---|---|---|---|---|---|
| (Intercept) | 2.6491 | 0.2927 | 2.6475 | 9.051 | 0.004601 | ** |
| Grana | −1.1644 | 0.1752 | 6 | −6.648 | 0.00056 | *** |
| Margin | −0.1012 | 0.1752 | 6 | −0.578 | 0.584522 | |
| Stroma | −1.4748 | 0.1752 | 6 | −8.42 | 0.000153 | *** |

*fnr1*
**Deletion test** carried out using Satterthwaite's method with the R package lmerTest (Kuznetsova, Brockhoff & Christensen 2017).
The model is a mixed effects model with random intercepts. The square root of response is the response variable, tissue is the fixed effect, and individual the random effect.

| Fixed effect deleted | Sum Sq | Mean Sq | Num DF | Den DF | F value | Pr (>F) | |
|---|---|---|---|---|---|---|---|
| Sub-compartment | 5.6516 | 1.8839 | 3 | 6 | 26.204 | 0.000759 | *** |

Model summary:

Random effects:

| Groups | Name | Variance | Std. Dev. | | | | |
|---|---|---|---|---|---|---|---|
| Individual | (Intercept) | 0.07521 | 0.2742 | | | | |
| Residual | | 0.07189 | 0.2681 | | | | |

Number of obs: 12, groups: individual, 3

Fixed effects when stroma is set as the intercept:

| | Estimate | Std. Error | DF | t value | Pr (>|t|) | |
|---|---|---|---|---|---|---|
| (Intercept) | 1.2117 | 0.2214 | 4.4836 | 5.472 | 0.003875 | ** |
| Grana | −0.1584 | 0.2189 | 6 | −0.724 | 0.496563 | |
| Lamellae | 0.9353 | 0.2189 | 6 | 4.272 | 0.005251 | ** |
| Margin | 1.5161 | 0.2189 | 6 | 6.925 | 0.000449 | *** |

*Table 2 continued on next page*

Fixed effects when lamellae is set as the intercept:

|  | Estimate | Std. Error | DF | t value | Pr (>|t|) |  |
|---|---|---|---|---|---|---|
| (Intercept) | 2.1469 | 0.2214 | 4.4836 | 9.695 | 0.000356 | *** |
| Grana | −1.0937 | 0.2189 | 6 | −4.996 | 0.002463 | ** |
| Margin | 0.5808 | 0.2189 | 6 | 2.653 | 0.037882 | * |
| Stroma | −0.9353 | 0.2189 | 6 | −4.272 | 0.005251 | ** |

*fnr1*:Zm*FNR1*
**Deletion test** carried out using Satterthwaite's method with the R package lmerTest (Kuznetsova, Brockhoff & Christensen 2017).
The model is a mixed effects model with random intercepts. The square root of response is the response variable, tissue is the fixed effect, and individual the random effect.

| Fixed effect deleted | Sum Sq | Mean Sq | Num DF | Den DF | F value | Pr (>F) |  |
|---|---|---|---|---|---|---|---|
| Sub-compartment | 4.5242 | 1.5081 | 3 | 6.01 | 23.558 | 0.001009 | ** |

Model summary:

Random effects:

| Groups | Name | Variance | Std. Dev. |
|---|---|---|---|
| Individual | (Intercept) | 0.0005984 | 0.02446 |
| Residual |  | 0.064017 | 0.25302 |

Number of obs: 12, groups: individual, 3

Fixed effects when stroma is set as the intercept:

|  | Estimate | Std. Error | DF | t value | Pr (>|t|) |  |
|---|---|---|---|---|---|---|
| (Intercept) | 1.1754 | 0.1468 | 7.9979 | 8.009 | $4.34^{-05}$ | *** |
| Relevel grana | 0.2318 | 0.2066 | 6.01 | 1.122 | 0.304712 |  |
| Relevel lamellae | 1.3774 | 0.2066 | 6.01 | 6.668 | 0.000547 | *** |
| Relevel margin | 1.2849 | 0.2066 | 6.01 | 6.22 | 0.000793 | *** |

Fixed effects when lamellae is set as the intercept:

|  | Estimate | Std. Error | DF | t value | Pr (>|t|) |  |
|---|---|---|---|---|---|---|
| (Intercept) | 2.55285 | 0.14676 | 7.9979 | 17.395 | $1.22^{-07}$ | *** |
| Relevel grana | −1.14567 | 0.20659 | 6.01002 | −5.546 | 0.001444 | ** |
| Relevel margin | −0.09252 | 0.20659 | 6.01002 | −0.448 | 0.669949 |  |
| Relevel stroma | −1.37744 | 0.20659 | 6.01002 | −6.668 | 0.000547 | *** |

*fnr1*:Zm*FNR2*

**Deletion test** carried out using Satterthwaite's method with the R package lmerTest (Kuznetsova, Brockhoff & Christensen 2017).
The model is a mixed effects model with random intercepts. The square root of response is the response variable, tissue is the fixed effect, and individual the random effect.

| Fixed effect deleted | Sum Sq | Mean Sq | Num DF | Den DF | F value | Pr (>F) |  |
|---|---|---|---|---|---|---|---|
| Sub-compartment | 5.4414 | 1.8138 | 3 | 6 | 22.849 | 0.001106 | ** |

Model summary:

Random effects:

| Groups | Name | Variance | Std. Dev. |
|---|---|---|---|
| Individual | (Intercept) | 0.02741 | 0.1656 |
| Residual |  | 0.07938 | 0.2817 |

Number of obs: 12, groups: individual, 3

Fixed effects when stroma is set as the intercept:

|  | Estimate | Std. Error | DF | t value | Pr (>|t|) |  |
|---|---|---|---|---|---|---|
| (Intercept) | 1.1886 | 0.1887 | 6.6796 | 6.3 | 0.000488 | *** |
| Grana | 0.4864 | 0.23 | 6 | 2.114 | 0.078885 | . |

*Table 2 continued on next page*

| | | | | | | |
|---|---|---|---|---|---|---|
| Lamellae | 1.8298 | 0.23 | 6 | 7.954 | 0.00021 | *** |
| Margin | 0.9239 | 0.23 | 6 | 4.016 | 0.006989 | ** |

Fixed effects when lamellae is set as the intercept:

| | Estimate | Std. Error | DF | t value | Pr (>\|t\|) | |
|---|---|---|---|---|---|---|
| (Intercept) | 3.0184 | 0.1887 | 6.6796 | 15.998 | $1.42^{-06}$ | *** |
| Grana | −1.3434 | 0.23 | 6 | −5.84 | 0.00111 | ** |
| Margin | −0.9059 | 0.23 | 6 | −3.938 | 0.00764 | ** |
| Stroma | −1.8298 | 0.23 | 6 | −7.954 | 0.00021 | *** |

*fnr1*:Zm*FNR3*

**Deletion test** carried out using Satterthwaite's method with the R package lmerTest (Kuznetsova, Brockhoff & Christensen 2017).
The model is a mixed effects model with random intercepts. The square root of response is the response variable, tissue is the fixed effect, and individual the random effect.

| Fixed effect deleted | Sum Sq | Mean Sq | Num DF | Den DF | F value | Pr (>F) | |
|---|---|---|---|---|---|---|---|
| Sub-compartment | 9.0046 | 3.0015 | 3 | 8 | 25.416 | 0.0001923 | *** |

Model summary:

Random effects:

| Groups | Name | Variance | Std. Dev. |
|---|---|---|---|
| Individual | (Intercept) | 0 | 0 |
| Residual | | 0.1181 | 0.3436 |

Number of obs: 12, groups: individual, 3

Fixed effects when stroma is set as the intercept:

| | Estimate | Std. Error | DF | t value | Pr (>\|t\|) | |
|---|---|---|---|---|---|---|
| (Intercept) | 1.06794 | 0.19841 | 8 | 5.383 | 0.00066 | *** |
| Grana | 0.06284 | 0.28059 | 8 | 0.224 | 0.828395 | |
| Lamellae | 1.99447 | 0.28059 | 8 | 7.108 | 0.000101 | *** |
| Margin | 1.44343 | 0.28059 | 8 | 5.144 | 0.00088 | *** |

Fixed effects when lamellae is set as the intercept:

| | Estimate | Std. Error | DF | t value | Pr (>\|t\|) | |
|---|---|---|---|---|---|---|
| (Intercept) | 3.0624 | 0.1984 | 8 | 15.435 | $3.09^{-07}$ | *** |
| Grana | −1.9316 | 0.2806 | 8 | −6.884 | 0.000127 | *** |
| Margin | −0.551 | 0.2806 | 8 | −1.964 | 0.085144 | . |
| Stroma | −1.9945 | 0.2806 | 8 | −7.108 | 0.000101 | *** |

associated with the algal thylakoid in Chlamydomonas (*Suss et al., 1995*), though they contradict an earlier study on higher plants (*Negi et al., 2008*), where the authors report co-localisation of FNR and GAPDH. In the chloroplast example shown by *Negi et al., 2008* some of the labelled FNR appears to be in the stroma, but no details of primary or secondary antibody specificity, sample number or leaf treatment prior to sample preparation are given, which would be necessary for a valid comparison with our work. It has previously been reported that FNR is a component of the inner envelope translocon (*Stengel et al., 2008*), but we detected little evidence for this in analysis of over 300 mature chloroplasts from 18 individuals (over 4500 gold particles counted). However, we cannot discount the possibility that such interactions are more prevalent earlier in chloroplast development, when protein import is more active.

*Benz et al., 2010* propose that FNR tethering by Tic62 is not involved in $NADP^+$ reduction. While these authors suggest that Tic62-binding prevents proteolytic degradation of FNR in the dark, the data in *Figure 5* reveal another function, critical for transient CEF on dark to light transition. This is consistent with earlier results in antisense tobacco plants, where transient CEF was negatively

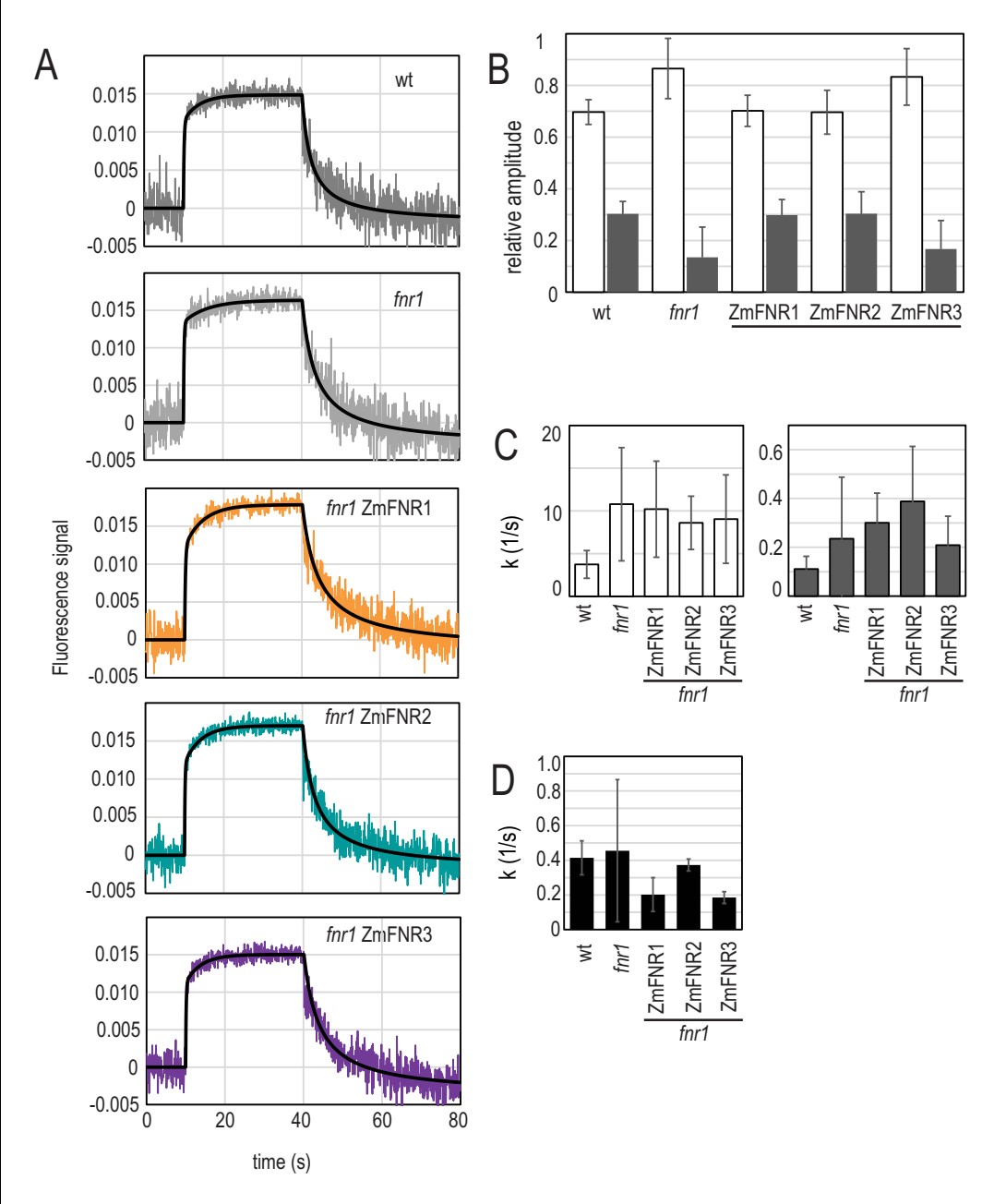

**Figure 4.** Light-dependent NADP(H) reduction and oxidation kinetics are influenced by FNR location and binding partners. (**A**) Traces show NADPH fluorescence of dark adapted Arabidopsis chloroplasts measured over a short light exposure from 10 to 40 s. Traces are averages of three to five independent chloroplast preparations, each of which was composed of an average of 15 separate measurements. Genotypes from which chloroplasts were isolated are indicated on each graph. Black traces overlaying signals are fits (two components for reduction, one component for oxidation), calculated as described in Materials and methods. (**B**) Relative amplitude of fast (white bars) and slow (grey bars) components fitted to the $NADP^+$ reduction traces shown in (**A**). (**C**) Rate of fluorescence induction for the fast (white bars, left) and slow (grey bars, right) components fitted to the $NADP^+$ reduction traces shown in (**A**). (**D**) Rate of NADPH fluorescence decay following switching off the light, fitted to the traces shown in A. **B–D** are averages of values calculated from three to five separate chloroplast preparations (parameters in *Supplementary file 2c and e*). All values given ± standard error (fitting errors with absolute, 95% confidence). See *Figure 4—figure supplement 1* for comparison between genotypes of proteins involved in NADP(H) metabolism. Appendix 1 describes further characterisation of the two phases of $NADP^+$ reduction and *Appendix 1—figure 1* shows further data on this topic.

*Figure 4 continued on next page*

*Figure 4 continued*

The online version of this article includes the following figure supplement(s) for figure 4:

**Figure supplement 1.** Abundance of the major photosynthetic complexes in the genotypes used in this study.

impacted by decreased FNR (*Joliot and Johnson, 2011*), although this effect has not previously been reported in Arabidopsis. Arabidopsis has two genes for FNR, and P700 re-reduction data from plants knocked-out or knocked-down for either gene have been interpreted as showing an increase in CEF after high light or temperature stress (*Hanke et al., 2008*; *Lintala et al., 2009*). FNR:Tic62 and FNR:TROL interactions are disrupted at higher light intensities (*Alte et al., 2010*; *Benz et al., 2009*), meaning that specific interaction to these tethers is unlikely to withstand illumination during stress treatments. Indeed, our ECS measurements after light acclimation show no difference in CEF between genotypes (*Figure 5A*), and P700 parameters indicate less donor limitation for *fnr1* than other genotypes (*Figure 5C*). The reported differences in P700 re-reduction (*Hanke et al., 2008*; *Lintala et al., 2009*) therefore likely have another cause in the *fnr1* mutants and *fnr2* knock-downs or reflect a different CEF pathway from the dark to light transition CEF measured in our work.

## The function of FNR bound to Tic62 and TROL

The study of CEF in full FNR knock-outs is hampered by the essential role FNR plays in generating NADPH in LEF (*Bendall and Manasse, 1995*; *Lintala et al., 2012*), and previous investigation of a role for FNR-location in CEF has been complicated by the decrease in total FNR content when the Tic62 or TROL proteins are knocked out (*Benz et al., 2009*; *Jurić et al., 2009*). Here we have generated plants with approximately wild type levels of FNR, but variable location (*Figure 2*), allowing us to establish that tight binding to Tic62 or TROL is necessary for the transient CEF associated with the dark light transition (*Figure 5A*). In fact, the literature investigating a role for FNR activity in CEF is extensive, with multiple inhibitor based studies supporting this hypothesis (*Ravenel et al., 1994*; *Shahak et al., 1981*; *Bendall and Manasse, 1995*; *Mills et al., 1979*; *Cleland and Bendall, 1992*; *Ye and Wang, 1997*; *Hosler and Yocum, 1985*). The actual function for FNR in higher plant CEF, however, remains speculative: it has been proposed as a direct Fd:quinone reductase (*Bojko et al., 2003*), or as a binding site for Fd to allow reduction of quinones via PgrL1 (*Hertle et al., 2013*) or the Cyt $b_6f$ in both algae and higher plants (*Buchert et al., 2018*; *Zhang et al., 2001*; *Buchert et al., 2020*). Although it is reported that interaction with Tic62 or TROL has no impact on FNR activity (*Alte et al., 2010*), another possibility is that by binding to Tic62 or TROL, FNR becomes less efficient at oxidising Fd. By default, reduced Fd would then donate electrons to other pathways, including CEF. This would be consistent with the recent finding that the NDH complex uses Fd as the direct electron donor (*Yamamoto and Shikanai, 2013*; *Schuller et al., 2019*). It has recently been reported that both the NDH and antimycin A CEF pathways are regulated by the ATP:ADP ratio (*Fisher et al., 2019*). In both cases ATP was found to act as a competitive inhibitor of reduced Fd association. It follows that an increase in the rate of Fd oxidation by FNR would result in greater ATP inhibition of CEF. Indeed, the *fnr1* and *fnr1-ZmFNR3* genotypes both show a greater fast-component of NADP$^+$ reduction (*Figure 4*), implying faster Fd oxidation at the onset of light, and both genotypes also lack upregulated CEF following dark adaptation (*Figure 5*).

The fact that transient CEF does not occur when total FNR content is decreased in the *fnr1* mutant (*Figure 5A*) and is decreased in *FNR* antisense tobacco (*Joliot and Johnson, 2011*) seems to support an active role for FNR, as any default Fd oxidation pathway would be expected to increase in these circumstances. Because several CEF pathways are regulated by redox signals (*Breyton et al., 2006*; *Takahashi et al., 2013*; *Strand et al., 2015*; *Strand et al., 2016*), it also remains possible that the impact of decreased FNR activity on CEF is indirect, through a failure to effectively poise the NADP(H) pool.

## FNR as a dynamic switch between LEF and CEF

Because the *fnr1* and *fnr1-ZmFNR3* genotypes, which lack Tic62 and TROL interaction (*Figure 2C*), retain FNR bound to the thylakoid (*Figure 3*), the enzyme must occupy alternative, weaker binding sites on the membrane following release from Tic62 and TROL. This is further supported by the IGL-staining on illuminated leaves (*Figure 6*), in which Tic62/TROL interactions with FNR should be

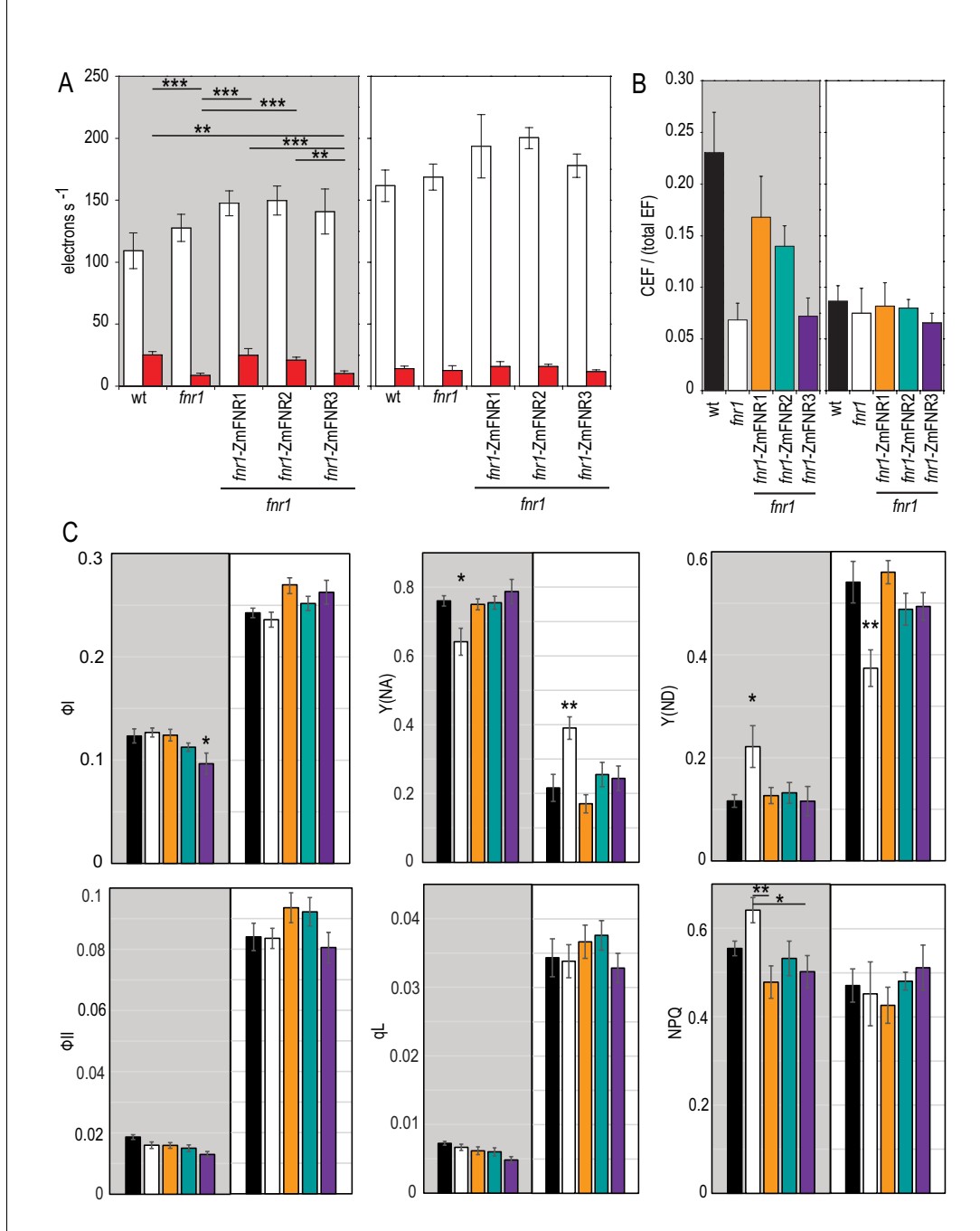

**Figure 5.** Impact of FNR location on photosynthetic electron transport in Arabidopsis following dark adaptation or light acclimation. (A) ECS measurements after a 20 s high light pulse on dark adapted leaves (grey background) and light acclimated leaves (5 min 150 μE m$^{-2}$ s$^{-1}$, actinic light). The relaxation kinetics of ECS were measured at 520–546 nm after a 20 s pulse of actinic light at 1100 μmol photons m$^{-2}$ s$^{-1}$ (LEF + CEF stimulating, white boxes) or far red light (CEF stimulating, red boxes). Averages are shown for between five and seven independent measurements ± s.d. (B) Relative amount of CEF as a function of total electron flow calculated from the data in (A). *Figure 5—figure supplement 1* shows a comparable experiment preformed with DCMU rather than far red light to drive PSI activity. (C) The response to high light in dark adapted leaves (grey background) and light acclimated leaves (5 min 150 μE m$^{-2}$ s$^{-1}$, actinic light, white background) was measured by detecting chlorophyll fluorescence and P700 oxidation with saturating pulses following 20 s of actinic light at 1100 μmol photons m$^{-2}$ s$^{-1}$. Measurements are averages ± s.e. of 6–12 replicates of the following genotypes: black, wt; white, *fnr1*; orange, *fnr1*-ZmFNR1; cyan, *fnr1*-ZmFNR2; purple, *fnr1*-ZmFNR3. Photosystem I parameters (quantum efficiency, ΦI; donor limitation, Y(ND); acceptor limitation, Y(NA)) and photosystem II parameters (quantum efficiency, ΦII; non-photochemical quenching, NPQ; photochemical quenching, qL). p-value significance in Tukey post hoc analysis of ANOVA is indicated as 0 '***', 0.001 '**' 0.01 '*' 0.05

*Figure 5 continued on next page*

*Figure 5 continued*
'.'. Unless indicated by bars, stars indicate variation from all other genotypes. Representative P700 and ECS traces are shown in *Figure 5—figure supplement 2*.

The online version of this article includes the following figure supplement(s) for figure 5:

**Figure supplement 1.** DCMU inhibition of PSII for measurement of cyclic electron flow in Arabidopsis.

**Figure supplement 2.** Raw data from ECS and PSI responses to high light treatment.

disrupted (*Alte et al., 2010*; *Benz et al., 2009*), but FNR remains membrane bound. We have no direct evidence for the identity of these weak sites of FNR-interaction, but it seems likely they include proteins that have previously been reported as FNR interaction partners, such as PSI (*Andersen et al., 1992*) or the $Cytb_6f$ (*Zhang et al., 2001*; *Clark et al., 1984*). Indeed, specific interaction between FNR and PSI has recently been demonstrated in vitro (*Marco et al., 2019*), and the authors suggest a binding site comprising PsaE and the light harvesting complexes. Proximity to PSI would result in more efficient flux of electrons from PSI to $NADP^+$, consistent with the dominant fast component of $NADP^+$ reduction in the *fnr1* mutant and *fnr1*-ZmFNR3 genotypes (*Figure 4*), where all FNR is weakly associated (*Lintala et al., 2007*; *Hanke et al., 2008*; *Figure 2*). The *fnr1* mutant is also the only genotype with higher FNR density at the margin than on the lamellae (*Figure 3*), consistent with the theory that LEF occurs predominantly on these regions of the thylakoid (*Anderson, 1992*; *Chow et al., 2005*). As the contribution of the slow component in $NADP^+$ reduction (*Figure 4*) matches the timescale of transient CEF on light to dark transition (*Joliot and Joliot, 2005*), and FNR interaction with Tic62 and TROL tethers is weakened in the light (*Alte et al., 2010*; *Benz et al., 2009*) we tentatively propose the following model to explain our data: in the dark, a large amount of FNR is sequestered at Tic62 and TROL on the lamellae in a location sub-optimal for Fd oxidation, limiting NADPH production before carbon fixation is upregulated. In this location FNR promotes CEF by an as yet uncharacterised mechanism, possibly related to Pgr5 and/or PgrL1. Over light induction, these interactions are disrupted, and the enzyme relocates to associate with PSI, resulting in increased efficiency of $NADP^+$ reduction. This speculative model is presented in *Figure 6—figure supplement 1*. Our measurements of $NADP^+$ reduction kinetics could be fitted by two components (*Figure 4A*, *Supplementary file 2c*), while this model would imply three possible changes in kinetics corresponding to dissociation, diffusion, and re-association. If the model in *Figure 6—figure supplement 1A* is correct, at least one component must be too fast to be detected. This could either be diffusion, due to close proximity of the Tic62/TROL membrane tethers to PSI, or association with PSI following a slow diffusion phase.

Although we propose that the slow phase of $NADP^+$ reduction kinetics (*Figure 4*) results from changes in FNR activity, we cannot exclude the possibility that it is caused by regulation of the rate of $NADP^+$ consumption. In this way, variable slow phase contributions could reflect differential regulation (originating in the redox state of the Fd and NADP(H) pools) of fast, NADPH consuming enzymatic processes downstream. The dominant NADPH consuming reaction in the chloroplast is catalysed by glyceraldehyde 3 P dehydrogenase (GAPDH) in the Calvin Benson Basham (CBB) cycle of $CO_2$ fixation, which is regulated by thioredoxin (Trx)-mediated thiol reduction and oxidised glutathione (*Wolosiuk and Buchanan, 1978*). Indeed, we previously measured differences in both steady state Trx regulation and glutathione redox poise associated with FNR abundance and membrane localisation (*Kozuleva et al., 2016*). Because thiol reduction of GAPDH acts to increase allosteric upregulation by 1,3 bisphosphoglycerate (*Baalmann et al., 1995*), and levels of CBB intermediates in isolated chloroplasts are very low, leading to a lag phase in starting the cycle (*Baldry et al., 1966*) it is unlikely that GAPDH activity will change significantly in dark adapted chloroplasts during the 30 s time-frame of our experiment. Nevertheless, it may be that even small changes in activity of such an abundant enzyme could contribute to the kinetics shown here. We also previously found that FNR abundance and location correlated with the thiol-mediated activation state of NADPH malate dehydrogenase (NADP-MDH) (*Kozuleva et al., 2016*), another powerful sink, which is upregulated much faster than GAPDH in isolated chloroplasts, as the allosteric regulator involved is NADPH itself (*Scheibe, 2004*). We therefore repeated the NADPH kinetic measurements with an *nadp-mdh* knock-out (*Appendix 1—figure 1*), but found no change in the amplitude of the slow phase of $NADP^+$ reduction relative to wild type. We cannot rule out an impact on other NADPH-dependent

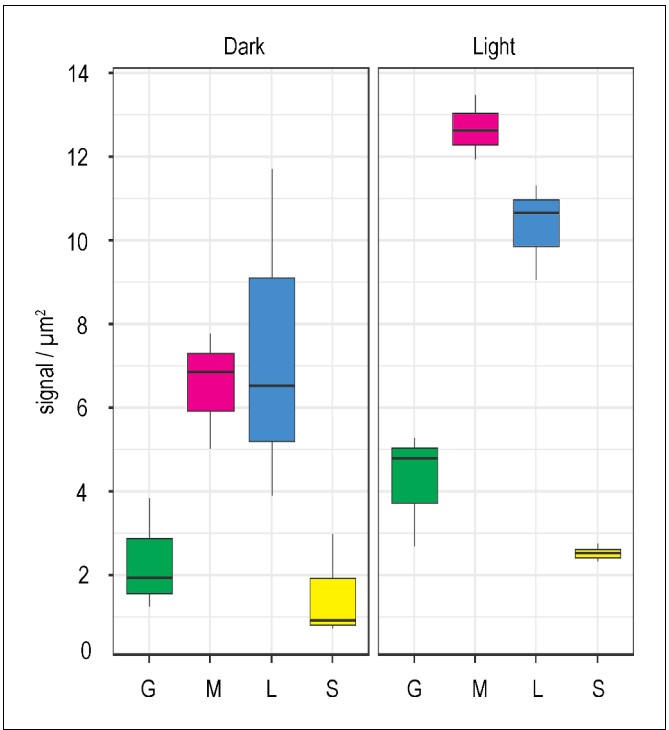

**Figure 6.** FNR sub-chloroplast distribution changes in response to light. Density of immunogold labelled FNR in different sub-chloroplast compartments of Wt Arabidopsis either dark incubated (left panel, same data as in *Figure 3*) or light incubated (right panel), prior to and during fixation. Densities given are grana core, green; margins, magenta; lamellae, blue; and stroma, yellow. Values are averages of three biological replicates, with combined label and area for the sub-compartments of 15–20 chloroplasts for each individual. Statistical significance within and between treatments was calculated with a mixed effects model analysis of variance (*Table 3*).

The online version of this article includes the following figure supplement(s) for figure 6:

**Figure supplement 1.** Two models describing potential mechanisms that could explain the impact of FNR interactions on photosynthetic electron transport during the dark to light transition.

processes in the chloroplast being influenced by FNR location, especially those regulated by thiol regulation cascades originating in the redox state of Fd or NADP(H) (*Yoshida and Hisabori, 2016*; *Naranjo et al., 2016*; *Nikkanen et al., 2016*; *Hashida et al., 2018*).

Based on the data shown in *Figure 5*, another intriguing possibility is an impact of CEF on NADP (H) turnover. For example, the slow phase of NADP$^+$ reduction might reflect activity of the proposed NADPH consuming CEF pathway utilising the reverse catalytic reaction of FNR when the enzyme is bound to the Cytb$_6$f (*Zhang et al., 2001*). Alternatively, upregulation of the NDH complex by NADPH poise (*Nikkanen et al., 2018*) or redox regulation (*Strand et al., 2015*) could be influenced by FNR location, resulting in differential competition for reduced Fd (*Schuller et al., 2019*) and variable substrate availability for FNR.

It is also possible that during the light FNR remains associated with the membrane tether proteins, but is activated through a change from strong to weak binding. We have interrogated this hypothesis by repeating the IGL experiment on Wt plants after light acclimation (*Figure 6*). We find that, in contrast to dark adapted plants, there is higher FNR density at the margins than on the lamellae (*Table 3*), supporting a model of some FNR diffusion to sites of high LEF. It therefore seems likely that, in higher plants, FNR does indeed change interaction partners and consequently subchloroplast location, as part of a mechanism to decrease CEF and increase LEF during light adaptation.

**Table 3.** Mixed effects model investigating changes in FNR density between different the chloroplast sub-compartments of WT Arabidopsis following dark adaptation or light adaptation of leaves.

Analysis performed using the data in *Figure 6*. Fixed effects taking either label density in the stroma as the intercept or label density in the margins/lamellae as the intercept. Linear mixed model fit by REML. Signif. codes: 0 '***' 0.001 '**' 0.01 '*' 0.05 '.' 0.1 '' 1.

**Deletion test** carried out using Satterthwaite's method with the R package lmerTest (Kuznetsova, Brockhoff & Christensen 2017). The model is a mixed effects model with random intercepts. The square root of response is the response variable, tissue is the fixed effect, and individual the random effect.

| Fixed effect deleted | Sum Sq | Mean Sq | Num DF | Den DF | F value | Pr (>F) | |
|---|---|---|---|---|---|---|---|
| Sub-compartment | 0.30184 | 0.10061 | 3 | 12 | 2.3613 | 0.1227 | |

Model summary:

Random effects:

| Groups | Name | Variance | Std. Dev. |
|---|---|---|---|
| Individual | (Intercept) | 0.1073 | 0.3276 |
| Residual | | 0.04261 | 0.2064 |

Number of obs: 24, groups: individual, 6

Fixed effects when dark adapted stroma is set as the intercept::

| | Estimate | Std. Error | DF | t value | Pr (>|t|) | |
|---|---|---|---|---|---|---|
| (Intercept) | 1.1743 | 0.2235 | 6.3067 | 5.253 | 0.00164 | ** |
| Grana | 0.3104 | 0.1685 | 12 | 1.842 | 0.09033 | . |
| Lamellae | 1.4748 | 0.1685 | 12 | 8.75 | $1.48^{-06}$ | *** |
| Margins | 1.3736 | 0.1685 | 12 | 8.15 | $3.11^{-06}$ | *** |
| Light:dark comparison | 0.4157 | 0.3161 | 6.3067 | 1.315 | 0.23427 | |
| Light:dark grana | 0.1421 | 0.2384 | 12 | 0.596 | 0.5622 | |
| Light:dark lamellae | 0.1482 | 0.2384 | 12 | 0.622 | 0.54562 | |
| Light:dark margins | 0.5963 | 0.2384 | 12 | 2.502 | 0.02782 | * |

Fixed effects when dark adapted lamellae is set as the intercept::

| | Estimate | Std. Error | DF | t value | Pr (>|t|) | |
|---|---|---|---|---|---|---|
| (Intercept) | 2.64913 | 0.223542 | 6.306738 | 11.851 | $1.52^{-05}$ | *** |
| Grana | −1.16437 | 0.168543 | 12.000001 | −6.908 | $1.63^{-05}$ | *** |
| Margins | −0.101175 | 0.168543 | 12.000001 | −0.6 | 0.5595 | |
| Stroma | −1.474807 | 0.168543 | 12.000001 | −8.75 | $1.48^{-06}$ | *** |
| Light:dark comparison | 0.563988 | 0.316136 | 6.306738 | 1.784 | 0.1223 | |
| Light:dark grana | −0.006164 | 0.238356 | 12.000001 | −0.026 | 0.9798 | |
| Light:dark margins | 0.44808 | 0.238356 | 12.000001 | 1.88 | 0.0846 | . |
| Light:dark stroma | −0.148241 | 0.238356 | 12.000001 | −0.622 | 0.5456 | |

Fixed effects when light acclimated stroma is set as the intercept:

| | Estimate | Std. Error | DF | t value | Pr (>|t|) | |
|---|---|---|---|---|---|---|
| (Intercept) | 1.5901 | 0.2235 | 6.3067 | 7.113 | 0.00031 | *** |
| Grana | 0.4525 | 0.1685 | 12 | 2.685 | 0.01986 | * |
| Lamellae | 1.623 | 0.1685 | 12 | 9.63 | $5.37^{-07}$ | *** |
| Margins | 1.97 | 0.1685 | 12 | 11.688 | $6.48^{-08}$ | *** |
| Dark:light comparison | −0.4157 | 0.3161 | 6.3067 | −1.315 | 0.23427 | |
| Dark:light grana | −0.1421 | 0.2384 | 12 | −0.596 | 0.5622 | |
| Dark:light lamellae | −0.1482 | 0.2384 | 12 | −0.622 | 0.54562 | |
| Dark:light margins | −0.5963 | 0.2384 | 12 | −2.502 | 0.02782 | * |

*Table 3 continued on next page*

*Table 3 continued*

**Deletion test** carried out using Satterthwaite's method with the R package lmerTest (Kuznetsova, Brockhoff & Christensen 2017). The model is a mixed effects model with random intercepts. The square root of response is the response variable, tissue is the fixed effect, and individual the random effect.

| Fixed effect deleted | Sum Sq | Mean Sq | Num DF | Den DF | F value | Pr (>F) | |
|---|---|---|---|---|---|---|---|
| Fixed effects when light acclimated lamellae is set as the intercept: | | | | | | | |
| | Estimate | Std. Error | DF | | t value | Pr (>\|t\|) | |
| (Intercept) | 3.213118 | 0.223542 | 6.306738 | | 14.374 | $4.68^{-06}$ | *** |
| Grana | −1.170535 | 0.168543 | 12.000001 | | −6.945 | $1.55^{-05}$ | *** |
| Margins | 0.346905 | 0.168543 | 12.000001 | | 2.058 | 0.062 | . |
| Stroma | −1.623048 | 0.168543 | 12.000001 | | −9.63 | $5.37^{-07}$ | *** |
| Dark:light comparison | −0.563988 | 0.316136 | 6.306738 | | −1.784 | 0.1223 | |
| Dark:light grana | 0.006164 | 0.238356 | 12.000001 | | 0.026 | 0.9798 | |
| Dark:light margins | −0.44808 | 0.238356 | 12.000001 | | −1.88 | 0.0846 | . |
| Dark:light stroma | 0.148241 | 0.238356 | 12.000001 | | 0.622 | 0.5456 | |

# Materials and methods

## Plant material and sample preparation

*Arabidopsis thaliana* plants were all Columbia ecotype. All transformed plants were screened at the level of western blotting (see later for Materials and methods) to confirm expression of heterologous FNR proteins prior to analysis. Plants were grown under a light/dark cycle of 12 h/12 hr with moderate light of 150 μmol photons $m^{-2}$ $s^{-1}$ at 22°C/18°C on soil. Samples were extracted from mature Arabidopsis leaves in the presence of 50 mM Tris-HCl, pH 7.5, 100 mM NaCl, 2 mM EDTA, 20 μg/ml, and 0.1 mg/ml polyvinyl-polypyrrolidone. Supernatant and pellet fractions were made from these extracts by centrifugation at 11,000 *g* 4°C for 5 min, and the membrane pellet was resuspended with an equal volume of buffer containing 0.1% Triton X-100 to solubilise proteins prior to analysis.

## Transmission electron microscopy

The transmission electron microscopy was performed on a Jeol JEM-1230 microscope (Jeol, Peabody, MA) equipped with a Morada CCD camera and iTEM Olympus software at 80.00 kV.

Immunogold labelling of leaf sections was carried out as follows. The first fully unfolded leaves of Arabidopsis Wt, *fnr1*, and plants expressing ZmFNR genes in the *fnr1* background were sampled for immunogold labelling to ensure consistency in developmental stage, and kept in the dark until the end of the fixation step. The leaves were harvested at the end of the dark period and cut into 1 mm strips with a sharp razor. The strips were transferred to a 3% paraformaldehyde/0.125 M phosphate buffered saline (PBS) in a syringe, creating an underpressure with the plunger to ensure full penetration of the tissue and removing any air from the parenchyma which would interfere with thin-sectioning later on. The fixation step lasted 5 min. For light adaptation, leaves were sampled 2 hr into a light period from under growth lights and maintained at 150 μE until fixation, which was also performed under light.

Leaves were embedded into LR White resin (Agar Scientific, Stansted) by sequential incubation in 70% EtOH 30 min, 90% EtOH 30 min, 100% EtOH 30 min, 100% EtOH 30 min, 50% EtOH/50% LR White 60 min, 100% LR White 60 min, and 100% LR White overnight. Embedded strips were transferred to gelatine capsules, filled to the top with LR White resin, and covered with a piece of wax. Hardening of the resin took place in an oven at 60°C for 2.5 hr. The capsule was removed and the resin block cleaned from the wax and subsequently used for thin-sectioning.

After cutting the blocks on a Reichert-Jung ultramicrotome (Leica, Nussloch, Germany) into 70 nm thin-sections, these were transferred onto EM nickel grids. These were then immunogold labelled by the following sequential incubations in a covered wet chamber: 50 μl 1.25 M PBS 2 min, 20 μl 5% $H_2O_2$ 5 min, 50 μl PBS/50 mM glycine 3 × 3 min, 5% BSA in PBS 10 min, 1:200 anti ZmFNR2 or anti Cyt *f* in 1% BSA in PBS 30 min, 50 μl 1% BSA in PBS 3 × 6 min, 10 μl 1:200 gold particle conjugated anti rabbit IgG in 1% BSA in PBS 30 min, 100 μl PBS 8 × 2 min, 50 μl 1%

glutaraldehyde 5 min, 100 µl $H_2O$ 8 × 2 min, 20 µl 4% uranyl acetate 4 min, 100 ml $H_2O$ 3 × 20 min. Following this the grids were air dried in a dust free container and ready to use in TEM.

The areas of interest on the electron micrographs were defined by printing at high resolution and manually colouring in magenta (margins), blue (lamellae), and green (grana core). See text for full explanation of chloroplast sub-compartment definitions in this work. Areas of chloroplasts with poor membrane resolution were not included in analysis. To account for antibody size, an area of 10 nm on either side of both margins and lamellae were included in the area (see *Figure 1—figure supplement 2*). These images were then scanned and analysed in ImageJ using the Versatile Wand Tool. By subtraction of the sum of grana, lamellae, and margin area from the total chloroplast area analysed, a value for the stroma was calculated. Then, the gold particles were manually counted on each micrograph (in the region of 10–50 per chloroplast) and the labelling density of each sub-compartment was calculated as particles/µm$^2$.

## Statistical analysis

Because of the presence of multiple measurements from the same individual, FNR densities were analysed using random-intercepts mixed effects models fitted using the lme4 (*Bates et al., 2015*) and lmerTest (*Kuznetsova et al., 2017*) packages in R version 1.1.456 (*R Development Core Team, 2019*). Sub-compartment (divided into Stroma, Grana, Cytosol and Margin/Lamellae, or Stroma, Grana, Margin, and Lamellae) was the fixed effect and individual the random effect. Following initial data exploration and model fitting the response variable was square root transformed for all models in order to reduce skew in the residuals. Models were fitted with both Stroma and Margin/lamellae or Lamellae set as the intercept in order to allow all important effects to be represented in the contrasts. For PAM and ECS analysis, the statistical analysis of variance (ANOVA) and post hoc analysis Tukey's tests were also performed in R.

## Generation of transgenic plants

Homozygous *fnr1* knock out Arabidopsis plants were transformed with a construct containing the maize FNR coding sequence as described previously (*Twachtmann et al., 2012*). The individual transformations only yielded positive results for the Zm*FNR2* construct and seeds of the second generation after transformation were used in this study. For generation of plants containing the maize coding sequence for FNR1 and FNR3, the *fnr1* mutant plant was crossed with Arabidopsis plants expressing Zm*FNR1* or Zm*FNR3*, respectively. The offspring seeds were selected for the expression of maize FNR genes on agar plates with the corresponding herbicide. For this study the third generation after transformation was used.

## Cloning and purification of maize TROL

RNA from sweetcorn-type 'Golden X Bantam' (Bingenheimer Saatgut AG, Echzell-Bingenheim, Germany) was isolated using TRI Reagent RT (Molecular Research Center Inc, Cincinnati, USA) according to the manufacturer's manual before reverse transcription into cDNA using the 'RevertAid H Minus First Strand cDNA Synthesis' kit (Thermo Fisher Scientific Inc, Waltham, USA). Coding sequence of maize TROL was amplified from this cDNA using primers 5'-GTCGACGAGGATCGACAAAA-3' and 5'-GAATTCCTAGACCCGGTTTCTT-3' containing restriction sites for *Sal* I and *EcoR*I, respectively. This product was ligated into pJET1.2/blunt (Thermo Fisher Scientific Inc, Waltham, USA) before subcloning with *Sal* I and *Xba*I into pCold-I (Takara Bio Inc, Shiga, Japan). Competent *Escherichia coli* BL21(DE3) cells were transformed with expression vector and positive transformants were selected on 100 µg/ml ampicillin. A single colony was used for inoculation of YT broth containing 50 µg/ml ampicillin and subsequent gene expression was performed according to the pCold-I manual. Harvested cells were resuspended in binding buffer containing 20 mM Tris-HCl pH 7.9, 500 mM NaCl, and 5 mM imidazole before addition of 1 mM Pefabloc SC (Carl Roth GmbH and Co. KG, Karlsruhe, Germany), 100 µg/ml lysozyme, and 0.1% Triton X-100. Cell suspension was lysed by sonication. Clarified lysate containing maize TROL-His(6x) was applied to a column containing chelating fast flow sepharose (GE Healthcare, Little Chalfont, England) previously charged with 50 mM $NiSO_4$ and equilibrated with binding buffer using the 'ÄKTAprime plus' FPLC system (GE Healthcare, Little Chalfont, England). The column was washed with the same buffer containing 52 mM imidazole before elution of bound protein with binding buffer containing a total of 300 mM imidazole.

Collected fractions were pooled and used for ammonium sulphate precipitation by adding ground powder up to 20% saturating solution. The solution was centrifuged for 10 min and 10,000 g at 4°C before resuspension of the obtained pellet in buffer containing 100 mM $NaHCO_3$ and 20 mM NaCl. The solution was dialysed against the same buffer overnight. In addition, the purified protein was dialysed with buffer containing 50 mM Tris-HCl pH 7.5 and 100 mM NaCl prior of generating primary antisera in guinea pig (Pineda Antikörper-Service, Berlin, Germany).

## Native and SDS-PAGE

SDS-PAGE samples have been treated with 0.5% SDS and 2.5% 2-mercaptoethanol and heated at 95°C for 5 min. Native-PAGE samples have been treated with 10% glycerol and 0.1% bromophenol blue before loading. Samples were separated by native-PAGE (*Kimata and Hase, 1989*) or SDS-PAGE (*Laemmli, 1970*) on a 12% gel before immunoblotting (*Towbin et al., 1979*) on polyvinylidene defluoride (PVDF) membrane (Bio-Rad Laboratories Inc, Hercules, USA). Antibodies used in immunoblotting detection were raised against PC (1:5000), LHCII (1:8000), LHCI (1:8000), PgrL1 (1:3000), cytochrome $b_6$ (1:50,000), NDHS (1:10,000), PsbA (1:10,000), PsaD (1:5000), spinach LHC (1:40,000) (all purchased from Agrisera, Vännäs, Sweden), maize (*Zea mays*) FNR2 (1:60,000), spinach (*Spinacea oleracea*) Cyt *f* (1:5000), and maize TROL (1:10,000).

## Blue native-PAGE

All steps were performed at 4°C. Chloroplasts have been isolated basically as described by *Hanke et al., 2008*. Samples have been prepared as described previously (*Twachtmann et al., 2012*) and been analysed by blue native-PAGE as described (*Reisinger and Eichacker, 2007*) using a 6% to 12% polyacrylamide gradient gel before immunoblotting on PVDF membrane (Bio-Rad Laboratories Inc, Hercules, USA) and detection of FNR and TROL by ECL with horseradish peroxidase. Antibodies used in immunoblotting detection were raised against maize FNR2 (1:50,000) and maize TROL (1:10,000).

## ΔNADPH fluorescence measurements

Chloroplasts were prepared essentially as described previously (*Hanke et al., 2008*), with the following modifications. All steps were performed in the dark at 4°C. Leaves of four to six developing plants per genotype (4–5 weeks old) were homogenised in a waring blender in 80 ml 25 mM Hepes-NaOH pH 8.0, 0.33 M sorbitol, 60 mM KCl, 10 mM EDTA, 1 mM $MgCl_2$, 0.4 mM ascorbate, 40 mg bovine serum albumin (BSA) before filtering through muslin and centrifugation for 2 min at 1150 *g* for 2 min. The pellet was resuspended in 2 ml of the same buffer before centrifuging at 736 *g* for 2 min. This time the pellet was resuspended in 1 ml of the same buffer before overlaying on a step gradient of 40% above 80% PB-Percoll (5% PEG 4000, 1% BSA in Percoll) in 0.33 M sorbitol, 5 mM Hepes-NaOH pH 7.5, 2 mM EDTA. After centrifuging at 3050 *g* for 1 min followed by no brake deceleration, intact chloroplasts were extracted from the boundary layer. Chloroplasts were washed twice in extraction buffer without ascorbate or BSA followed by centrifugation at 1700 *g* and resuspension of the pellet in 200–500 µl extraction buffer without ascorbate or BSA. Between 10 and 20 µg chlorophyll were used in each assay of ΔNADPH fluorescence.

ΔNADPH fluorescence in chloroplast suspensions was performed basically as described for cyanobacteria in *Kauny and Sétif, 2014* with the following adaptations. Fluorescence was measured in a volume of 3 ml extraction buffer at 25°C using the NADPH/99-A module of a Dual-PAM. Chloroplasts were dark adapted throughout preparation (about 40 min) and not stirred during the measurement. Gain was set low and damping was set high, while in the dark the measuring frequency was set to 100 Hz, changing to 5000 Hz on illumination. The measuring light intensity was set at 4 (corresponding to 9 µmol photons $m^{-2}$ $s^{-1}$). Actinic illumination was with red light from the Dual-PAM at maximum intensity, corresponding to approximately 750 µmol photons $m^{-2}$ $s^{-1}$ in the cuvette. Data were collected using the clock cycle function to trigger a run of 10 s dark, 30 s actinic light, 40 s dark recovery. Data shown are averages of 7–15 traces, which were manually checked for anomalies and to ensure that the first and last spectra collected did not differ significantly. Spectra presented are corrected for the constant drift that occurred during illumination. Further information on characterisation of this technique is given in Appendix 1.

## Fitting of ΔNADPH kinetics

The increase in fluorescence (NADP$^+$ reduction) under illumination is clearly bi-phasic and we fit these kinetics to a model which assumes a first order fluorescence increase kinetics with a fast and slow component,

$$F_{decay}(t) = F_\infty \left[ A_{slow} \left( 1 - e^{-k_{slow}t} \right) + A_{fast} \left( 1 - e^{-k_{fast}t} \right) \right]$$

where $F_\infty$ is the asymptotic level of fluorescence in the long time limit. The relative amplitudes $A_{slow}$ and $A_{fast}$ are not independent but related by,

$$A_{slow} + A_{fast} = 1$$

meaning the fit parameters are the rate constants $k_{slow}$ and $k_{fast}$, $A_{slow}$ and $F_\infty$. No improvement to the fit was obtained by adding further components.

The fluorescence decrease (NADPH oxidation) appears to follow simple second-order kinetics. Initially, for generality, Hill type kinetics were assumed,

$$F_{rec}(t) = \frac{F_\tau - F'_\infty}{1 + (k_{rec}(t - \tau))^n} + F'_\infty$$

where $\tau$ is the time at which the actinic illumination ceased, $F_\tau$ is the fluorescence level at that time, $F'_\infty$ is the asymptotic level of fluorescence and $n$. Fixing $n = 1$ (non-cooperative or true second order kinetics) did not alter the fit but significantly reduced the errors on the remaining fit parameters ($F'_\infty$ and $k_{rec}$).

## Cyclic electron flow measurements

Plants were grown under long day conditions and dark incubated for 30 min before transfer into actinic light with 150 µE m$^{-2}$ s$^{-1}$ PAR for light acclimation. After 20 s or 5 min of illumination, LEF and CEF were measured by following the relaxation kinetics of the carotenoid electrochromic bandshift at 520 nm (corrected for the signal at 546 nm) using a JTS-10 spectrophotometer (Biologic, France). The ECS spectral change is a shift in the pigment absorption bands that is linearly correlated with the light-induced generation of a membrane potential across the thylakoid membranes (*Bailleul et al., 2010*). Under steady state continuous illumination, the ECS signal stems from transmembrane potential generation by PSII, the cytochrome $b_6 f$ complex, and PSI and from transmembrane potential dissipation by the ATP synthase CF$_0$-F$_1$. When light is switched off, reaction centre's activity stops immediately, while ATPase and the cytochrome $b_6 f$ complex activities remain (transiently) unchanged. Therefore, the initial rate of ECS decay is proportional to the rate of PSI and PSII photochemistry (i.e. to the rate of 'total' electron flow). This can be calculated dividing this rate (expressed as -ΔI/I per unit of time) by the amplitude of the ECS signal (again expressed as −ΔI/I) induced by the transfer of one charge across the membrane (e.g. one PSI turnover). The rate of CEF can be evaluated using the same approach under conditions where PSII activity is inhibited. Typically, this is done by preventing PSII activity with DCMU. In our experiments, we found that leaf infiltration with DCMU did not result in homogeneous inhibition of PSII, as tested by measuring fluorescence transients in the leaves using an imaging setup (Speedzen, JbeamBio, France), causing high variability and high estimations of CEF (*Figure 5—figure supplement 1*). Therefore, another approach was employed to evaluate CEF, i.e. exposure to saturating far red light (λ >720 nm), to fully excite PSI with a minimum excitation of PSII. Results were expressed as electrons$^{-1}$ s$^{-1}$ and estimated from the amplitude of the electrochromic shift signal upon excitation with a saturating single turnover flash (five ns laser pulse). Total electron flow was measured following a pulse of actinic light (λ = 640 ± 20 nm FWHM) at 1100 µmol photons m$^{-2}$ s$^{-1}$ while CEF was measured with a pulse of far red light at the maximum setting (estimated as 1400 µmol photons m$^{-2}$ s$^{-1}$ by the manufacturer).

## Chlorophyll fluorescence and P$_{700}$ absorption

Chlorophyll fluorescence and P$_{700}$ absorption were simultaneously analysed using the DUAL-PAM-100 system (Heinz Walz GmbH, Effeltrich, Germany), and parameters calculated as described in the DUAL-PAM handbook (*Klughammer and Schreiber, 2008*). Plants were first dark incubated for 30 min before determination of F$_0$ as dark fluorescence yield. Change in P700 signal was monitored as

the difference in transmittance at 875 nm and 830 nm. The maximum change in $P_{700}$ signal indicated as $P_m$ was measured by application of far red light for 30 s followed by application of a saturating pulse. We confirmed that $P_m$ was not significantly underestimated in mutants with decreased acceptor limitation at PSI (*Supplementary file 2f*). In addition the maximum fluorescence yield $F_m$ of dark adapted leaves was determined before illumination. For the dark adapted measurement, plants were directly subjected to high light (1100 µE m$^{-2}$ s$^{-1}$ PAR) with saturation pulses applied after 5 s and 20 s. Light acclimated plants were instead illuminated in the DUAL-PAM leaf clip at 150 µE m$^{-2}$ s$^{-1}$ PAR for 5 min before application of the 1100 µE m$^{-2}$ s$^{-1}$ high light and saturating pulses after 5 s and 20 s. Typical fluorescence and P700 oxidation traces are shown in *Figure 5—figure supplement 2*. Only data from the pulses applied after 20 s of high light were used to generate the data presented. The maximal fluorescence yield $F_m'$ or maximum change in $P_{700}$ signal $P_m'$ were determined on each saturating pulse, while the current fluorescence yield F was measured before each pulse.

## Accession numbers

Arabidopsis Genome Initiative locus identifiers for the genes mentioned in this article are as follows: maize *FNR1*, BAA88236; maize *FNR2*, BAA88237; maize *FNR3*, ACF85815; Arabidopsis *FNR1*, AT5G66190; Arabidopsis *FNR2*, AT1G20020; maize Tic62, ACG28394.1; Arabidopsis *Tic62*, AT3G18890; maize *TROL*, ACF79627.1; Arabidopsis *TROL*, AT4G01050.1.

## Acknowledgements

This work was supported by funding from the Deutsche Forschungsgemeinschaft through (Project 2 in the Collaborative Research Center (SFB) 944) and BBSRC (BB/R004838/1) to G.T.H. M.K. was partly supported by a Geoff Schell fellowship from the Bayer Science and Education foundation (F-2016-BS-0555). G.F. acknowledges funds from the INRA, and the HFSP and the support of the GRAL LabEX GRAL, ANR-10-LABX-49–01 financed within the University Grenoble Alpes graduate school (Ecoles Universitaires de Recherche) CBH-EUR-GS (ANR-17-EURE-0003). Support from the European Research Council: ERC Chloro-mito (grant no. 833184) is also acknowledged. A.K.L. benefits from the support of the LabEx Saclay Plant Sciences-SPS (ANR-10-LABX-0040-SPS) and the French Infrastructure for Integrated Structural Biology (FRISBI) ANR-10-INSB-05. We thank Giulia Mastroianni and Kirsten Jäger for excellent technical support and Pierre Setif, Renate Scheibe, Michael Hippler and Toshiharu Hase for helpful discussions. Finally, we acknowledge the initial advice and encouragement regarding statistical analysis of IGL from Steve LeComber (QMUL), who sadly passed away during preparation of the manuscript. The authors wish to dedicate this paper to the memory of Professor Giorgio Forti, who was a pioneer in the study of FNR in vascular plants.

## Additional information

### Funding

| Funder | Grant reference number | Author |
|---|---|---|
| Deutsche Forschungsgemeinschaft | SFB944 project 2 | Manuela Kramer<br>Guy Thomas Hanke |
| Biotechnology and Biological Sciences Research Council | BB/R004838/1 | Guy Thomas Hanke |
| University Grenoble Alps | ANR-10-LABX-49-01 | Giovanni Finazzi |
| H2020 European Research Council | 833184 | Giovanni Finazzi |
| LabEx Saclay Plant Sciences-SPS | ANR-10-LABX-0040-SPS | Anja Krieger-Liszkay |
| French Infrastructure for Integrated Structural Biology | ANR-10-INSB-05 | Anja Krieger-Liszkay |
| Bayer CropScience | F-2016-BS-0555 | Manuela Kramer |
| University Grenoble Alps | CBH-EUR-GS (ANR-17- | Giovanni Finazzi |

EURE-0003)

The funders had no role in study design, data collection and interpretation, or the decision to submit the work for publication.

## Author contributions

Manuela Kramer, Formal analysis, Investigation, Methodology, Writing - review and editing; Melvin Rodriguez-Heredia, Laura Mosebach, Investigation; Francesco Saccon, Formal analysis, Investigation; Manuel Twachtmann, Investigation, Writing - review and editing; Anja Krieger-Liszkay, Investigation, Methodology, Writing - review and editing; Chris Duffy, Formal analysis, Methodology; Robert J Knell, Data curation, Formal analysis; Giovanni Finazzi, Conceptualization, Investigation, Methodology, Writing - review and editing; Guy Thomas Hanke, Conceptualization, Supervision, Funding acquisition, Investigation, Writing - original draft, Project administration

## Author ORCIDs

Robert J Knell http://orcid.org/0000-0002-3446-8715
Guy Thomas Hanke https://orcid.org/0000-0002-6167-926X

## Decision letter and Author response

Decision letter https://doi.org/10.7554/eLife.56088.sa1
Author response https://doi.org/10.7554/eLife.56088.sa2

# Additional files

## Supplementary files

• Supplementary file 1. Summary of published information about Arabidopsis and maize FNR isoproteins. Taken from *Hanke et al., 2005*, *Okutani et al., 2005*, *Lintala et al., 2009*, *Lintala et al., 2007*, and *Twachtmann et al., 2012*. Kinetic parameters are for the reverse direction to photosynthesis (NADPH-dependent reduction of Fd).

• Supplementary file 2. Additional statistical analysis. (a) Table of mixed effects model investigating changes in FNR density between different chloroplast sub-compartments in WT Arabidopsis. Analysis of data presented in *Figure 1—figure supplement 2*. Fixed effects taking either label density in the stroma as the intercept or label density in the margins/lamellae as the intercept. Linear mixed model fit by REML. Signif. codes: 0 '***' 0.001 '**' 0.01 '*' 0.05 '.' 0.1 ' ' 1. (b) Table of mixed effects model investigating changes in cytochrome $f$ density between different chloroplast sub-compartments in WT Arabidopsis. Analysis of data presented in *Figure 1—figure supplement 2*. Fixed effects taking either label density in the stroma as the intercept or label density in the margins/lamellae as the intercept. Linear mixed model fit by REML. Signif. codes: 0 '***' 0.001 '**' 0.01 '*' 0.05 '.' 0.1 ' ' 1. (c) Table of fitting parameters and errors in comparison of light-dependent $NADP^+$ reduction by different genotypes. Analysis performed using the data in *Figure 4*. Fits were calculated from experiments on individual chloroplast preparations, and then the parameters, and the fitting errors averaged. (d) Table of statistical analysis on the contribution of the fast phase to total amplitude of light-dependent fluorescence change in the chloroplast assay of $NADP^+$ reduction. Analysis performed using the data averaged in *Figure 4* and in (c). (e) Table of fitting parameters and errors in comparison of dark NADPH oxidation by different genotypes. Analysis performed using the data in *Figure 4*. Fits were calculated from experiments on individual chloroplast preparations, and then the parameters, and the fitting errors averaged. (f) Table of Pm values and statistical analysis of plants used for PAM analysis of the high light response. Analysis performed using the data in *Figure 5*, with example traces given in *Figure 5—figure supplement 2*. Pm determination of dark adapted leaves in order to calculate PSI parameters in response to high light treatment of Wt, *fnr1*, and *fnr1* plants expressing either ZmFNR1, ZmFNR2, or ZmFNR3 Arabidopsis plants (see *Figure 5*). n = 5–7 replicates. Signif. codes: 0 '***' 0.001 '**' 0.01 '*' 0.05 '.' 0.1 ' ' 1.

• Transparent reporting form

## Data availability

All data generated or analysed during this study are included in the manuscript and supporting files, except individual electron micrographs. Micrographs of chloroplasts that were analysed have been deposited at Dryad (https://doi.org/10.5061/dryad.7d7wm37rs) for full transparency. These are marked, to indicate the areas of the chloroplast analysed.

The following dataset was generated:

| Author(s) | Year | Dataset title | Dataset URL | Database and Identifier |
|---|---|---|---|---|
| Kramer M, Hanke GT | 2020 | Immunogold labelling of chloroplasts | https://doi.org/10.5061/dryad.7d7wm37rs | Dryad Digital Repository, 10.5061/dryad.7d7wm37rs |

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

## Appendix 1

### Characterisation and interpretation of the 2 phases fit to NADP(H) reduction and oxidation kinetics in intact plastids

Several phases of light-dependent fluorescence in chloroplasts (most much faster than those described here) at equivalent wavelengths have previously been described and extensive characterisation indicates that they are due to light-dependent reduction of $NADP^+$, with little or no contribution from protein content (*Latouche et al., 2000*). Nevertheless, as the relative amplitude of these two $NADP^+$ reduction phases (*Figure 4*) is dependent on tight binding of FNR to the thylakoid (*Figure 2*), we sought to further characterise our system.

We initially assumed that only intact chloroplasts were likely to contribute to the signal. This is because *Latouche et al., 2000* estimated that as little as 2.5–3% of all NADP(H) is free in the stroma (amounting to approximately 1–3 µM), meaning soluble NADP(H) from ruptured chloroplasts would be subject to a huge dilution factor at the concentrations in our assay (chlorophyll concentrations of 3–6 µg /ml). However, controls with deliberately ruptured chloroplasts give some light-dependent kinetics (*Appendix 1—figure 1*), albeit at lower amplitude and with higher noise than intact chloroplasts. One explanation for this could be that a significant proportion of chloroplast NADP(H) remains bound strongly to the FNR enzyme. It is estimated that as much as 44% of chloroplast NADPH is bound to FNR (*Latouche et al., 2000*). If NADP+ remains bound to FNR in a solution of ruptured chloroplasts, then a cascade of PSI > Fd > FNR could support its reduction. When data from ruptured chloroplast traces are fitted, they do not reveal distinct phases, but rather a rapid reduction and oxidation. This demonstrates that ruptured chloroplasts are not the source of the slow phase of $NADP^+$ reduction.

Although NADH levels are reported to be undetectably low in isolated chloroplasts (*Heineke et al., 1991*) light-dependent $NAD^+$ reduction in leaf chloroplasts was recently reported using fluorescent markers (*Lim et al., 2020*) and might contribute specifically to one of the phases detected in our measurements. FNR affinity for $NAD^+$ is extremely low (*Piubelli et al., 2000*) but indirect flux through other enzymes is possible, the most likely route being an internal malate valve comprising both NADP(H)- and NAD(H)-dependent malate dehydrogenase (MDH) enzymes (*Selinski and Scheibe, 2019*). To test this possibility, the experiment was repeated with a *nadp-mdh* mutant, which shows identical kinetics to the wt (*Appendix 1—figure 1*), indicating that light-dependent $NAD^+$ reduction probably makes an undetectable contribution to the signal we measure.

These results and the work of *Latouche et al., 2000* indicate that the kinetics detected in *Figure 4* and *Appendix 1—figure 1* result from light-dependent changes in the turnover of NADP(H) in intact chloroplasts. As the measurements were performed on dark adapted chloroplasts, the slow phase likely reflects the relatively slow (10–20 s) light-dependent regulation of either a fast NADP(H) reduction or oxidation process. Because the amplitude of the slow phase correlates with FNR:FNR-tether protein interaction (*Figure 2*), and its time frame corresponds to the generation of ΔpH (shown by *Alte et al., 2010* to drive dissociation of FNR from tether proteins), we hypothesise that this could correspond to upregulation of $NADP^+$ reduction, on release of FNR from the tether proteins. However, it is also possible that the slow phase originates in light-dependent changes to downstream NADPH consumption processes, and it may be that secondary effects of FNR-FNR tether protein interactions also have an impact here. For example, this might particularly be the case for processes subject to thiol-regulation by cascades originating in the redox state of NADP(H) (*Yoshida and Hisabori, 2016*; *Naranjo et al., 2016*; *Nikkanen et al., 2016*; *Hashida et al., 2018*). Another possibility might be slow saturation of an NADPH-dependent electron sink, the most likely candidate being glutathione reductase (*Foyer and Halliwell, 1976*). However, we have found that, compared to the Wt, glutathione is in a more oxidised state in *fnr1* plants (*Kozuleva et al., 2016*), which would not be consistent with the smaller contribution of the slow phase to $NADP^+$ reduction kinetics in this genotype.

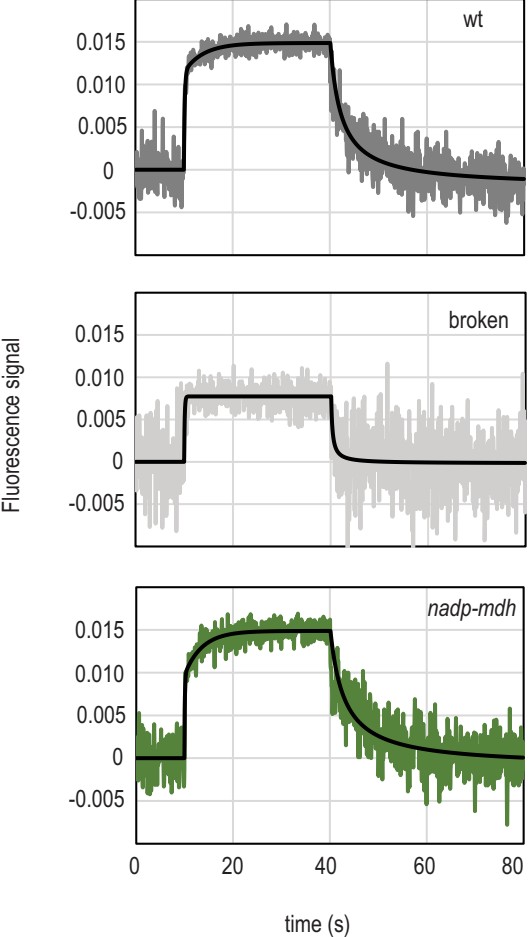

**Appendix 1—figure 1.** Examination of the two phases detected in fluorescent measurement of light-dependent NADP$^+$ reduction. Traces show NADPH fluorescence of dark adapted Arabidopsis chloroplasts measured over a short light exposure from 10 to 40 s. Traces are averages of five separate chloroplast preparations (wt), or representative of two separate experiments (broken chloroplasts and the *nadp-mdh* mutant, each of which is composed of an average of 15 separate measurements). Genotypes and treatment of chloroplasts are indicated on each graph. Chloroplasts were ruptured either by freeze thawing, or osmotic lysis with the same result. Black traces overlaying signals are fits (two components for reduction, one component for oxidation), calculated as described in Materials and methods.

