## [Decision Letter]

**Acceptance summary:**

The work makes significant contribution to our understanding of photosynthesis, specifically in the reactions following transfer of electrons from photosystem I, and on how they are partitioned among downstream sinks (metabolic processes) and cyclic electron flow, an important process that has received relatively little experimental attention. The reviewers thought that the revised version has been substantially improved by more robust analyses and consideration of alternative interpretations of the results, setting up the field to directly test and validate the mechanisms.

**Decision letter after peer review:**

Thank you for submitting your article "Electron flow during photosynthesis is regulated by location of Ferredoxin:NADP(H) Oxidoreductase" for consideration by *eLife*. Your article has been reviewed by two peer reviewers, one of whom is a member of our Board of Reviewing Editors, and the evaluation has been overseen by Christian Hardtke as the Senior Editor. The following individual involved in review of your submission has agreed to reveal their identity: Robert Burnap (Reviewer #2).

The reviewers have discussed the reviews with one another and the Reviewing Editor has drafted this decision to help you prepare a revised submission.

As the editors have judged that your manuscript is of interest, but as described below that additional experiments are required before it is published, we would like to draw your attention to changes in our revision policy that we have made in response to COVID-19 (https://elifesciences.org/articles/57162). First, because many researchers have temporarily lost access to the labs, we will give authors as much time as they need to submit revised manuscripts. We are also offering, if you choose, to post the manuscript to bioRxiv (if it is not already there) along with this decision letter and a formal designation that the manuscript is 'in revision at *eLife*'. Please let us know if you would like to pursue this option. (If your work is more suitable for medRxiv, you will need to post the preprint yourself, as the mechanisms for us to do so are still in development.)

Summary:

This work describes significant, new results that help answer long-standing question about the roles of FNR in regulating the partitioning of electrons from photosystem I (PSI) into linear electron flow (LEF) and cyclic electron flow (CEF). Both reviewers felt that the work makes a strong contribution to a long-standing and perplexing area of photosynthesis research. One strength of the work is that it critically tests some of the past assumptions about how, or even if, this re-localization occurs, and comes to the provocative conclusion that much of the previous work needs to be re-evaluated, with important consequences for a number of models proposed in the literature. Specifically, the previously proposed soluble forms of FNR are shown to remain membrane-associated, but that the localization within different thylakoid domains is variable. They then perform a number of genetics and spectroscopic measurements that are interpreted to support a role for the changes in localization of FNR to regulating electron flow. However, both reviewers also concluded that there were substantial issues, in experimental interpretation and statistical analysis.

Essential revisions:

Both reviewers also concluded that there were substantial issues that must be addressed. Some of these issues may require some additional experimental work, but it is also possible that they can be dealt with by more rigorous analysis and interpretation of the results the group already has on hand. For example, it may be possible to deal with the reviewer #1's criticism of the interpretation of photosystem I absorbance signals, by reanalyzing the raw data.

However, other issues, e.g. the statistical analysis in Figure 4 brought up by both reviewers, appear to require additional experimental results, and this may delay resubmission.

Both reviewers recommended specific approached to most easily address these issues.

The discussion should be modified along the lines of the reviewers' comments to better reflect the strengths and weaknesses of the data sets and the current state of the field.

While it is understandable that one would as high impact a title as possible, it should not make declarations that are not supported by the paper itself.

Reviewer #1:

This manuscript revisits a long-standing, and quite mysterious, set of observations in which FNR1 and FNR2 appear can be differentially localized near the thylakoid membrane or in the stroma, possibly regulating the partitioning of electrons from photosystem I (PSI) into linear electron flow (LEF) and cyclic electron flow (CEF). What I like about the work is that it critically reviews some of the past assumptions about how, or even if, this re-localization occurs, and comes to the provocative conclusion that much of the previous work needs to be re-evaluated, with important consequences for a number of models proposed in the literature. Specifically, the previously proposed soluble forms of FNR are shown to remain membrane-associated, but that the localization within different thylakoid domains is variable. They then perform a number of genetics and spectroscopic measurements that are interpreted to support a role for the changes in localization of FNR to regulating electron flow.

While some of the work seems to be solid, there are serious experimental issues with three areas of photosynthetic measurements and their interpretation.

Critical science issues:

The interpretation of the data in Figure 4 is likely to be over simplistic, or at the very least, there are multiple interpretations that lead to different conclusions.

The test interprets the fast and slow rise phases as reflecting more rapidly or more slowly interacting FNR/NADPH pools, as in the following: "We therefore propose that the slow component reflects relocation of FNR from tightly bound tether locations to weak associations elsewhere on the thylakoid."

However, this seems unlikely considering the slow time scale of the slow phases. More likely explanations include the following:

1) It is also possible that the system is heterogeneous, so that different subpopulations of chloroplasts or pools within chloroplasts behave differently. For instance, the presence of a fraction of broken or damaged chloroplasts could give rise to a slow NADP+ reduction phase;

2) Reduction of other electron carriers, including NADH which would also contribute to blue fluorescence, or bound NAD(P)H which has been reported to have a higher fluorescence yield than their free forms;

3) The slow rise represents is controlled by downstream re-oxidation of previously reduced NADPH. The immediate rise in fluorescence is, indeed, consistent with a model in which a local pool of NADPH goes rapidly reduced when the light is switched on. In this case, The slow rise may reflect NADPH further net reduction of NADPH, but not because of low accessibility of FNR/NADP+ but rather changes in the reoxidation of reduced NADPH. In this case, in initial extent of NADPH reduction is set by competition between reduction and re-oxidation, and the slow rise is caused by either increased in reduction or decreases in re-oxidation. For example, there may be a pool of oxidant for NADPH that is slowly reduced during the slow phase. Or, a local pool of O2 could be depleted during the slow rise time.

The complex, multiphasic decay kinetics in the dark seem to be consistent with this type of model. For example, there are clearly fast phases of decay in fnr1 and fnr1 ZmFNR2 that are not fit to the "simple second order kinetics" model used in the manuscript. (Indeed, there is no strong rationale for using simple second-order kinetics, which assumes that the system is homogeneous but with limiting concentrations of two substrates.) This simple second order model cannot account for the rapid decay phases (see above). For instance, it is possible that a very rapid re-oxidation process would introduce a quite rapid decay phase that, even if small in amplitude, would constitute a large fraction of the flux. Indeed, the kinetics are quite complex and thus simple second order behavior is likely to be an oversimplification. This may also explain the improved fits when modulating the ad hoc Hill cooperativity term.

“We then determined that these differences in NADP(H) kinetics are not due to secondary impact on other PET chain components, by examining abundance of proteins that could impact photosynthetic electron transport or NADP(H) poise (Figure 4—figure supplement 1). We found no differences that correlate to those in NADP(H) reduction and oxidation kinetics.”

This is not a very strong argument. Even with equal content there could be different activities or locations of other enzymes, as is inferred from the other arguments in the manuscript.

"In both cases this yields a linear relationship. It is reported that FNR interactions with Tic62 and TROL are weakened by exposure to light (Alte et al., 2010, Benz et al., 2009)."

The lack of statistical treatment in this data set is troubling. In Figure 4, the error bars shown in bar graphs are apparently not related to biological or even technical replicates, but to "fitting errors", presumably from a single trace. This issue extends into Figure 5, so it is not possible to conclude that there is a linear relationship or not. The apparent linearity is mostly due to a single point, fnr1, and removing this point would suggest that the relationship is quite different, perhaps an exponential rise or no statistically significant relationship at all.

As described previously for pea 235 chloroplasts (Schreiber and Klughammer, 2009) and Arabidopsis (Hanke et al., 2008), 9 isolated chloroplasts show distinct components of NADP+ 236 reduction.

"We then assessed the impact of FNR location on photosynthetic electron transport…"

This part of the text makes a rather big leap in inferences from a hypothetical interpretation of blue fluorescence kinetics to the location of the FNR…

“In order to differentiate LEF from CEF, we compared white light illumination (stimulation of both PSI and PSII, and therefore CEF + LEF), with far red light illumination (PSI excitation only, and therefore CEF only).”

This is clearly incorrect. While far red light preferentially excites PSI, it does not exclusively do so, and thus it is not possible to attribute all electron transport or associated signals to CEF under far red light. Also, the previous observation of high CEF during the first 20 s of illumination with light hitting both PSI and PSII were attributed to metabolic or physiological effects of the induction of downstream metabolic processes. It is not at all clear if the same would be true with far red light, nor that it would be comparable in the mutant lines.

"Unexpectedly, after dark adaptation the fnr1 mutant, which theoretically has lower Fd oxidation capacity, shows lower acceptor limitation than the other genotypes."

There are several potential issues with these results.

No references are given to the methods.

The predominant method for making these types of measurements will likely not work in these types of mutants and conditions. Neither the text nor the Materials and methods supply sufficient information to determine important aspects of the experiment, nor are representative traces given, so it is not possible to determine if this is the case.

It is important to show the kinetics of the P700 signals because with strong PSI acceptor limitations a strong pulse of actinic light can result in rapid accumulation of electrons on PSI acceptors, preventing one from determining the real amplitude of Pm or Pm'. In many cases, application of far red light minimized this issue, but in the case of fnr1, in which the capacity for oxidation of Fd is constitutively low, it is quite likely that this will not help. Underestimating Pm should artifactually decrease Y(NA).

The best way to determine if these issues impact the results is to show the raw traces and look for the transient behavior of the P700+ signal.

"The non-photochemical quenching parameters for PSI were calculated with Y(ND) = 1-P700 red. and Y(NA) = (Pm-Pm') / Pm. The effective PSI quantum yield was calculated with Y(I) = 1-Y(ND)-Y(NA). NPQ was calculated by the DUAL-PAM-100 software."

These are not "non-photochemical quenching" parameters, but relative changes in redox states.

"Chloroplasts switch rapidly between LEF and CEF..". One of the best demonstration of the rapid rates of switching between LEF and CEF is Luker et al., who showed it was rapid indeed, on the order of tens of seconds. How can changing the localization of FNR and its interactions with the thylakoid membrane respond this quickly?

Reviewer #2:

This is a strong contribution that analyzes the correlation between the distribution of FNR alone sections of the chloroplast and the characteristics of cyclic electron flow. The topic addresses important controversial question about how cyclic electron transport is transiently upregulated and them down regulated during the dark light transition ---- a brief period when excess light energy cannot yet be adequately utilized because the time it takes to activate the carbon fixation reactions of the Calvin-Benson-Bassham cycle. Significance also relates to The fact that similar regulatory features are probably necessary for the protection stabilization of the photosynthetic mechanism in fluctuating environmental conditions. Therefore, I rate the overall significance this manuscript as high.

Technically, this appears to be a well-executed set of wisely chosen experiments. There are some grammatical issues, but overall the manuscript is clearly written with interesting discussions. Apart from some clarifications on the interpretation of the kinetics, I find no serious glitches in this article and offer only a few minor additional suggestions:

It would be a good idea to briefly define what control coefficient means since this aspect of metabolic theory is not so widely studied, despite its importance.

Results paragraph seven (this paragraph and next):

This discussion would benefit from some re-working to clarify. After finishing the Discussion I can see what they were after, but the initial description of the kinetic features needs clarification. Later on a graphical model would also help.

Figure 4 is fit with two components, yet the discussion would seem to imply a third sequential component that needs to be accounted for in the fit. If the strongly bound form is not simply "less efficient", but essentially inactive, then slow component refers to the time of diffusion and (or release assuming very fast diffusion). In that case, the third component could be essentially invisible, if the proposed diffusional excursion ends with much faster process. Intermediate situations would be a more complex situation and I think it would probably appear as a more sigmoidal characteristic of the slow phase as those mobilized convert to faster activity. Don't know if this makes sense, but definitely the authors need to better express their ideas in the context of a model.

I think this is not a change in efficiency but the existence of two parallel reactions with different efficiencies (see above).

The authors should consider the recent finding that the NADH-1 complex appears not to use NADPH, but Fd which would be in line with their interpretation.

[Editors' note: further revisions were suggested prior to acceptance, as described below.]

Thank you for resubmitting your work entitled "Regulation of photosynthetic electron flow on dark to light transition by Ferredoxin:NADP(H) Oxidoreductase interactions" for further consideration by *eLife*. Your revised article has been evaluated by Christian Hardtke (Senior Editor) and a Reviewing Editor.

The manuscript has been improved but there are some remaining issues that need to be addressed before acceptance, as outlined below:

Essential revisions:

The reviewers judged that the current revisions corrected most of the issues, mainly by eliminating some areas of over-interpretation and narrowing the focus of the discuss. However, some important problems remain, especially regarding the interpretation of the NADPH florescence kinetics. As brought up by reviewer #1 in the first review, and further expanded on be reviewer #2 in the second review, the biphasic characteristics of the fluorescence rise can have multiple interpretations, and this issue has not been addressed in the revised text, which still sticks to an over-simplistic interpretation and the text needs to introduce other possible interpretations especially because the data presented thus far cannot exclude them. The reviewer and editor feel that an acknowledgement and full discussion of these possibilities is not only essential for the science, it will strengthen the paper substantially.

The paper may be acceptable for publication if this, and the smaller issues brought up by Reviewer #2, can be addressed. In particular, the revision must address the inaccurate interpretation of the Kauny/Setif reference and discuss possible alternative interpretations of the fast and slow phases of the NADPH kinetics.

Reviewer #2

Although the regulation of cyclic electron flow (CEF) in plant chloroplasts has been clearly demonstrated, its mechanism has yet to be established. The present work investigates the hypothesis that regulation of CEF is exerted via the re-positioning of ferredoxin-NADP reductase (FNR) to different places on the thlyakoid. The work begins with an immunogold electron microscopic analysis of the distribution of FNR in chloroplast of several Arabidopsis mutants that have previously shown to have different associations with the thylakoids, though interestingly, all forms appear to be associated with the thylakoid and none free in the stroma. Correlations between the distribution of FNR and the characteristics of CEF was then anlayzed using multiple approaches, and although some assays were not necessarily unambiguous, overall the data presents a reasonably compelling argument that FNR re-positioning underlies the observed switching in the intensity of CEF, at least during the dark-light transition.

By and large the questions have been mostly addressed, however some problems remain regarding the NADPH florescence kinetics, specifically their biphasic characteristics of their rise. The authors still have not excluded other downstream processes as being responsible for the slow phase. These can be slowly changing fast rates of redox sinks. In other words, these can be relatively fast NADPH consumption process (very fast relative to the times scales of the experiment), but their rates can be slowly changing. The argument post about the lack of biphasicity cyanobacteria (Kauny/Setif reference) is not accurate: although clearly different kinetics, with different shapes are involved in cyanobacteria, the slower phases are clearly visible and actually correlate to things like state transitions and the activation of the Calvin cycle. Accordingly, processes downstream of PSET (but interacting due to back-propagation effects) seem to be responsible for these multi second kinetic features. Since thee are intact chloroplasts, it is clearly possible such effects are present in the current experiments. Thus the argument based on measuring the abundance of electron transport chain components is not sufficient to strongly argue against the dominance of downstream processes in governing the slower phase. This needs to be clearly articulated in the main results text as well as in discussion. That said, the hypothesis that relocation accounts for these phases is not excluded either and altogether the interpretations seem to hold together. Nevertheless, the cited weakness needs to temper the strength of the arguments.

---

## [Author Response]

Essential revisions:Both reviewers also concluded that there were substantial issues that must be addressed. Some of these issues may require some additional experimental work, but it is also possible that they can be dealt with by more rigorous analysis and interpretation of the results the group already has on hand. For example, it may be possible to deal with the reviewer #1's criticism of the interpretation of photosystem I absorbance signals, by reanalyzing the raw data.However, other issues, e.g. the statistical analysis in Figure 4 brought up by both reviewers, appear to require additional experimental results, and this may delay resubmission.

We have now completed this experimental work, which is presented in a new Figure 4 and Appendix 1—figure 1.

Both reviewers recommended specific approached to most easily address these issues.The discussion should be modified along the lines of the reviewers' comments to better reflect the strengths and weaknesses of the data sets and the current state of the field.

The discussion has been modified according to the reviewers recommendations.

While it is understandable that one would as high impact a title as possible, it should not make declarations that are not supported by the paper itself.

The title has been modified to more specifically reflect the findings in the paper.

Reviewer #1:This manuscript revisits a long-standing, and quite mysterious, set of observations in which FNR1 and FNR2 appear can be differentially localized near the thylakoid membrane or in the stroma, possibly regulating the partitioning of electrons from photosystem I (PSI) into linear electron flow (LEF) and cyclic electron flow (CEF). What I like about the work is that it critically reviews some of the past assumptions about how, or even if, this re-localization occurs, and comes to the provocative conclusion that much of the previous work needs to be re-evaluated, with important consequences for a number of models proposed in the literature. Specifically, the previously proposed soluble forms of FNR are shown to remain membrane-associated, but that the localization within different thylakoid domains is variable. They then perform a number of genetics and spectroscopic measurements that are interpreted to support a role for the changes in localization of FNR to regulating electron flow.While some of the work seems to be solid, there are serious experimental issues with three areas of photosynthetic measurements and their interpretation.Critical science issues:The interpretation of the data in Figure 4 is likely to be over simplistic, or at the very least, there are multiple interpretations that lead to different conclusions.

We thank the reviewer for pushing us toward a more rigorous analysis. Chloroplast preparations can be highly variable depending on growth conditions, etc. We repeated the experiment several times, even in different laboratories prior to our first submission, always finding the same basic trends (principally the absence or small contribution of the slow phase in the fnr1 mutant). Because of the variation between experiments performed on different days and in different labs, we selected a representative dataset (with the clearest signal) for fitting. We have now repeated the experiment a further 3-6 times for each genotype with plants grown in identical conditions to obtain robust averages, and performed statistical analysis, showing that differences between genotypes are not random. We present these data as a new Figure 4. Several of the phenomena we described in the previous submission are not visible in the individual replicates and are therefore not statistically significant when averaged. These are therefore no longer discussed, and the text has been amended accordingly.

The test interprets the fast and slow rise phases as reflecting more rapidly or more slowly interacting FNR/NADPH pools, as in the following: "We therefore propose that the slow component reflects relocation of FNR from tightly bound tether locations to weak associations elsewhere on the thylakoid."However, this seems unlikely considering the slow time scale of the slow phases. More likely explanations include the following:

We agree that all the suggestions raised by the reviewer are possible, and that it is important to consider them. However, with respect to reviewer 1 we still believe that our suggestion is (at present) more likely for the reasons stated in the answers to individual points below. We have included a new section (Appendix 1). detailing further characterisation of the NADP+ reduction kinetics to test the hypotheses presented by the reviewer to explain the presence of these two phases.

1) It is also possible that the system is heterogeneous, so that different subpopulations of chloroplasts or pools within chloroplasts behave differently. For instance, the presence of a fraction of broken or damaged chloroplasts could give rise to a slow NADP+ reduction phase;

All chloroplast preparations are by their nature heterogeneous, and we mention in the text that this is a reason for not comparing total amplitudes. We have performed the experiment with deliberately ruptured chloroplasts (Appendix 1—figure 1) which show a considerably lower light dependent signal, greater noise and the absence of the slow kinetic phase. As we now write in the text (Appendix 1), we think that the reason for the residual light dependent activity is that a considerable proportion of chloroplast NADP(H) is thought to be bound to the FNR enzyme (Latouche et al., 2000), and therefore can be reduced effectively even in a ruptured chloroplast system. This experiment does establish that the slow phase of kinetics is dependent on intact chloroplasts.

On the other hand we certainly agree with the reviewer that different pools of NADP(H) within chloroplasts may behave differently.

2) Reduction of other electron carriers, including NADH which would also contribute to blue fluorescence, or bound NAD(P)H which has been reported to have a higher fluorescence yield than their free forms;

Indeed, protein bound NADPH has a higher fluorescence than NADPH free in solution (L.N.M Duysens, G.H.M Kronenberg (1957) BBA, 26: 437-438; T.G Scott, R.D Spencer, N.L Leonard, G Weber (1970) JACS, 92:687-695), and so we initially thought that the slow phase we measure might be due to slow association of NADPH with enzymes. FNR binds a considerable proportion of free NADP+ (Latouche et al., 2000) and it could therefore be that the decreased FNR abundance in the fnr1 mutant might be related to the decreased amplitude of the slow phase. However, heterologous expression of ZmFNR3 in the fnr1 mutant yields Wt levels of FNR, but still fails to rescue the amplitude of this slow phase, suggesting that FNR location, rather than abundance is responsible. This is in line with the findings of Latouche et al. (2000), who conclude that neither protein binding, nor flavins within enzymes (an important consideration when using plants with variable FNR content) contribute to light dependent NADPH fluorescence in isolated chloroplasts after extensive characterisation of the technique, as we now discuss in Appendix 1.

On conducting these experiments we assumed negligible contribution of NADH to the fluorescence, based on the undetectably low NADH levels reported in illuminated, isolated chloroplasts by (Heineke et al., 1991, Plant Physiology 95, 1131). However, since our intial submission, there has been a report that, based on expression of NADPH and NADH responsive fluorescent proteins in cytosol and stroma, NADH levels do indeed rise in the stroma at the onset of light (Lim et al., Nat Comm 11, 3238). FNR affinity for NAD(H) is extremely low (Piubelli et al., 2000) and no enzyme capable of light dependent reduction of NAD^+^ has yet been identified. NADPH dependent reduction of NAD^+^ is possible, as suggested by Lim et al. (2020) and we have tested this in the following way. The dominant pathway for NADPH/NADH interconversion is an internal chloroplast OAA/malate shuttle, comprising NADP(H) malate dehydrogenase and NAD(H) malate dehydrogenase (both present in the chloroplast, see Selinsky and Scheibe 2018). Thiol upregulation of NADP-MDH over a similar time scale as the development of the slow phase makes this a good candidate. We therefore repeated our NADP(H) kinetics experiment with a T-DNA knock out of this enzyme (nadp-mdh), of which an example trace is shown in Appendix 1—figure 1. If light dependent NAD^+^ reduction is contributing to the fluorescence seen in our measurement, then this mutant should show significant perturbation in the kinetics, but instead it shows a nearly identical kinetic pattern to the Wt. This is now discussed in Appendix 1. In our opinion, this means a contribution of NADH to the light dependent kinetics in our chloroplast preparations is unlikely.

3) The slow rise represents is controlled by downstream re-oxidation of previously reduced NADPH. The immediate rise in fluorescence is, indeed, consistent with a model in which a local pool of NADPH goes rapidly reduced when the light is switched on. In this case, The slow rise may reflect NADPH further net reduction of NADPH, but not because of low accessibility of FNR/NADP+ but rather changes in the reoxidation of reduced NADPH. In this case, in initial extent of NADPH reduction is set by competition between reduction and re-oxidation, and the slow rise is caused by either increased in reduction or decreases in re-oxidation. For example, there may be a pool of oxidant for NADPH that is slowly reduced during the slow phase. Or, a local pool of O2 could be depleted during the slow rise time.

If we interpret this comment correctly, the reviewer suggests that the slow phase in Figure 4 could be due to an increase in the rate of NADP^+^ reduction, or a decrease in the rate NADPH oxidation over the “slow phase”. In the first case – this is consistent with what we are suggesting in the paper. We now provide 2 models (Figure 6—figure supplement 1) showing possible ways that NADP^+^ reduction activity and CEF might change according to FNR location over the dark to light transition. We base these models on the fact that the slow phase amplitude correlates with genotypes that have strong FNR: membrane interaction, and that in Wt these interactions are reported to be disrupted in the light (Benz et al., 2009) and at pH values expected in the stroma during the light (Alte et al., 2010).

In the second case, it is possible that, as the reviewer suggests, there could be a decrease in the rate of NADPH consumption. We now mention this in Appendix 1. The reviewer suggests as an example the oxidation of a localised pool of O_2_. Although it is hard to understand how this would correlate with tight binding of FNR to Tic62/TROL tethers, we have considered that it could be due to slow saturation of glutathione reduction (by NADPH dependent glutathione reductase), which would be indirectly connected to O_2_ reduction. However, in previous work we found that under illumination the glutathione pool of the fnr1 mutant is actually more oxidised than that of the Wt (Kozuleva et al., 2016), which would be at odds with the smaller contribution of the slow phase seen in this mutant in Figure 4. As the data we present in Figure 4 are averages of repetitive measurements on isolated chloroplasts, it is unlikely that it represents exhaustion of a metabolite resulting in a decrease in NADPH oxidation.

Finally, the slow phase seen in our (and other) chloroplast preparations is absent from NADPH fluorescence measurements on cyanobacteria (Kauny and Setif, 2014). In cyanobacteria, FNR is permanently bound to the thylakoid by a phycobilisome linker peptide (Korn et al. 2009, JBC 284, 31789), so movement between complexes / locations is not likely. This indicates that the slow phase of NADP^+^ reduction is either related to a specific property of higher plant FNR (such as interaction with the Tic62/TROL membrane tethers and various other complexes) or chloroplast specific metabolism. This is now also discussed in Appendix 1. We accept that our explanation that the slow phase may be due to FNR dissociating from one complex and re-associating with another (as previously suggested by Johnson and Joliot, among others) could prove to be wrong, but based on current evidence it seems the most likely explanation in our opinion.

The complex, multiphasic decay kinetics in the dark seem to be consistent with this type of model. For example, there are clearly fast phases of decay in fnr1 and fnr1 ZmFNR2 that are not fit to the "simple second order kinetics" model used in the manuscript. (Indeed, there is no strong rationale for using simple second-order kinetics, which assumes that the system is homogeneous but with limiting concentrations of two substrates.) This simple second order model cannot account for the rapid decay phases (see above). For instance, it is possible that a very rapid re-oxidation process would introduce a quite rapid decay phase that, even if small in amplitude, would constitute a large fraction of the flux. Indeed, the kinetics are quite complex and thus simple second order behavior is likely to be an oversimplification. This may also explain the improved fits when modulating the ad hoc Hill cooperativity term.

The differences in NADPH oxidation could not be clearly discerned in all the individual chloroplast preparations, and so we have removed this analysis and any discussion of it from the text. Regarding the Hill coefficient, we acknowledge that the fitting is purely phenomenological and we make no attempt to fit a physical model to the data – it is simply used to obtain some useful characterization of the kinetics. The parameters probably have little physical meaning in and of themselves but in a relative sense they are informative. We chose a Hill type function because neither mono- nor bi-exponential functions provided a reasonable fit. Adding further exponentials would probably improve the fit of the data, but also run the danger of over-fitting.

“We then determined that these differences in NADP(H) kinetics are not due to secondary impact on other PET chain components, by examining abundance of proteins that could impact photosynthetic electron transport or NADP(H) poise (Figure 4—figure supplement 1). We found no differences that correlate to those in NADP(H) reduction and oxidation kinetics.”This is not a very strong argument. Even with equal content there could be different activities or locations of other enzymes, as is inferred from the other arguments in the manuscript.

The reviewer is right, there could be variation in PET regulation between the genotypes, but establishing that the transgenic approach used has not resulted in changes to the abundance of relevant proteins is a first step to eliminating pleiotropic effects as a factor. If there are regulatory differences, it is more likely that they are at least a response to different FNR activity / location (such as the redox state of Fd or NADP(H)), which we do mention in the text, as pointed out by the reviewer. Nevertheless, we have now changed the text to clarify that protein content does not always equate to activity.

"In both cases this yields a linear relationship. It is reported that FNR interactions with Tic62 and TROL are weakened by exposure to light (Alte et al., 2010, 262 Benz et al., 2009)."The lack of statistical treatment in this data set is troubling. In Figure 4, the error bars shown in bar graphs are apparently not related to biological or even technical replicates, but to "fitting errors", presumably from a single trace. This issue extends into Figure 5, so it is not possible to conclude that there is a linear relationship or not. The apparent linearity is mostly due to a single point, fnr1, and removing this point would suggest that the relationship is quite different, perhaps an exponential rise or no statistically significant relationship at all. Overall, even though we

Data from our multiple chloroplast preps was not consistent enough for us to draw these conclusions, and so we have removed the former Figure 5, and discussion of it.

As described previously for pea 235 chloroplasts (Schreiber and Klughammer, 2009) and Arabidopsis (Hanke et al., 2008), 9 isolated chloroplasts show distinct components of NADP+ 236 reduction."We then assessed the impact of FNR location on photosynthetic electron transport…"This part of the text makes a rather big leap in inferences from a hypothetical interpretation of blue fluorescence kinetics to the location of the FNR…

If the reviewers is suggesting that the hypothesis that FNR is relocated within the chloroplast is too big a leap, it has previously been proposed in published, peer reviewed work (Joliot and Johnson, 2011). Given the topic of our studies this published hypothesis cannot be easily ignored. This is even more the case given the correlation between NADP(H) fluorescence and FNR location seen when analysing the plants used in this work. The hypothesis is then tested in the experiment shown in Figure 6 later in the manuscript.

“In order to differentiate LEF from CEF, we compared white light illumination (stimulation of both PSI and PSII, and therefore CEF + LEF), with far red light illumination (PSI excitation only, and therefore CEF only).”This is clearly incorrect. While far red light preferentially excites PSI, it does not exclusively do so, and thus it is not possible to attribute all electron transport or associated signals to CEF under far red light. Also, the previous observation of high CEF during the first 20 s of illumination with light hitting both PSI and PSII were attributed to metabolic or physiological effects of the induction of downstream metabolic processes. It is not at all clear if the same would be true with far red light, nor that it would be comparable in the mutant lines.

We draw the reviewers attention to Figure 5—figure supplement 1, showing the same trend (although with high variation) when using DCMU rather than far red light to differentially stimulate PSII. In addition, as stated in the text (Materials and methods and legend of Figure 5) the illumination period was not under far red light, but actinic light, ensuring that the two treatments are equivalent. The only time far red light is used is for a 20 second period before the ECS measurement. We have modified the text to clarify this further. We do not suggest that any downstream effects are induced during the 20 s illumination, the function of which is to stimulate maximum activity of either both photosystems (actinic light) or preferentially stimulate PSI (far red light). It has previously been published that CEF can be equally assessed from ECS with far red light (as in our work), or Post Illumination Fluorescence Transients (PIFT) (Suorsa et al. (2016). Molecular plant, 9, 271-288).

"Unexpectedly, after dark adaptation the fnr1 mutant, which theoretically has lower Fd oxidation capacity, shows lower acceptor limitation than the other genotypes."There are several potential issues with these results.No references are given to the methods.

We have expanded the text in the Materials and methods to explain the methods in more detail, and referenced the Dual-PAM handbook, from where all calculations are taken, rather than setting out the calculations themselves.

The predominant method for making these types of measurements will likely not work in these types of mutants and conditions. Neither the text nor the Materials and methods supply sufficient information to determine important aspects of the experiment, nor are representative traces given, so it is not possible to determine if this is the case.

We have now cited the source of the calculations, and provided more information about the treatment conditions. We draw the reviewers attention to the representative P700 traces given in Figure 5—figure supplement 2. This was previously mentioned only in the figure legends, but we have now also mentioned it in the Materials and methods to make this clearer.

It is important to show the kinetics of the P700 signals because with strong PSI acceptor limitations a strong pulse of actinic light can result in rapid accumulation of electrons on PSI acceptors, preventing one from determining the real amplitude of Pm or Pm'. In many cases, application of far red light minimized this issue, but in the case of fnr1, in which the capacity for oxidation of Fd is constitutively low, it is quite likely that this will not help. Underestimating Pm should artifactually decrease Y(NA).

We did indeed check that our data were not due to underestimation of Pm, but did not include this data in the original manuscript. These are now supplied as a table in Supplementary file 2f, and show a small (but not significant) decrease in the original Pm measurement for the fnr1 mutant, indicating that this is not a significant problem in our measurement. This is now mentioned in the text. In addition, it should be pointed out that following light adaptation, the Y(NA) value of the fnr1 mutant becomes significantly higher (about 200%) than that of other genotypes (Figure 5), implying that the lower value seen after dark adaptation is not an underestimation.

The best way to determine if these issues impact the results is to show the raw traces and look for the transient behavior of the P700+ signal.

See earlier comment above about Figure 5—figure supplement 2.

"The non-photochemical quenching parameters for PSI were calculated with Y(ND) = 1-P700 red. and Y(NA) = (Pm-Pm') / Pm. The effective PSI quantum yield was calculated with Y(I) = 1-Y(ND)-Y(NA). NPQ was calculated by the DUAL-PAM-100 software."These are not "non-photochemical quenching" parameters, but relative changes in redox states.

This typo has been removed, along with this section of text, which has been replaced by a reference. Many thanks for drawing our attention to it.

"Chloroplasts switch rapidly between LEF and CEF..". One of the best demonstration of the rapid rates of switching between LEF and CEF is Luker et al., who showed it was rapid indeed, on the order of tens of seconds. How can changing the localization of FNR and its interactions with the thylakoid membrane respond this quickly?

We think that FNR relocation is definitely not the only mechanism of regulating CEF/LEF pathways, and probably not the dominant one – especially in high light. It is therefore hard to speculate about which CEF pathway is being measured in the work by Luker et al. based on our findings. However, we can elaborate on the suggestion we make in the discussion: FNR interactions with TROL and Tic62 tethers are subject to pH regulation (Alte et al., 2010), weakening as pH rises in the stroma. The change in Δ pH at the onset of light happens on the timescale of seconds or tens of seconds, providing a possible prompt for FNR release. In terms of association with (presumably) PSI and antennae (as shown by Marco et al., 2019), nothing is known about regulation, but given the failure to detect it in blue native gels it must be weaker than with TROL or Tic62. In the absence of other evidence to date, our assumption is that this interaction takes place by default on release from Tic62/TROL on a timescale of tens of seconds. We hope to examine this in future work.

Reviewer #2:This is a strong contribution that analyzes the correlation between the distribution of FNR alone sections of the chloroplast and the characteristics of cyclic electron flow. The topic addresses important controversial question about how cyclic electron transport is transiently upregulated and them down regulated during the dark light transition ---- a brief period when excess light energy cannot yet be adequately utilized because the time it takes to activate the carbon fixation reactions of the Calvin-Benson-Bassham cycle. Significance also relates to The fact that similar regulatory features are probably necessary for the protection stabilization of the photosynthetic mechanism in fluctuating environmental conditions. Therefore, I rate the overall significance this manuscript as high.Technically, this appears to be a well-executed set of wisely chosen experiments. There are some grammatical issues, but overall the manuscript is clearly written with interesting discussions. Apart from some clarifications on the interpretation of the kinetics, I find no serious glitches in this article and offer only a few minor additional suggestions:It would be a good idea to briefly define what control coefficient means since this aspect of metabolic theory is not so widely studied, despite its importance.

We have included a definition in the text.

Results paragraph seven (this paragraph and next):This discussion would benefit from some re-working to clarify. After finishing the Discussion I can see what they were after, but the initial description of the kinetic features needs clarification. Later on a graphical model would also help.

The paragraph, and a section of the discussion have been re-written to try and improve clarity. We are reluctant to include a model as a main figure, as we believe this is still quite speculative, but we now present two possible alternative graphical models as an additional Figure in Figure 6—figure supplement 1.

Figure 4 is fit with two components, yet the discussion would seem to imply a third sequential component that needs to be accounted for in the fit. If the strongly bound form is not simply "less efficient", but essentially inactive, then slow component refers to the time of diffusion and (or release assuming very fast diffusion). In that case, the third component could be essentially invisible, if the proposed diffusional excursion ends with much faster process. Intermediate situations would be a more complex situation and I think it would probably appear as a more sigmoidal characteristic of the slow phase as those mobilized convert to faster activity. Don't know if this makes sense, but definitely the authors need to better express their ideas in the context of a model.

This is an excellent point, and we thank the reviewer for the insightful comment. We have amended the text about dissociation, and removed the speculation about weak associations from the Results section, and now integrate it into the Discussion, amended to try and better express that i) dissociation, diffusion and association could be expected to contribute 3 distinct components, ii) that one of these components is either silent (possibly because of inactivation on binding), or too fast to be detected.

I think this is not a change in efficiency but the existence of two parallel reactions with different efficiencies (see above).

We agree with the reviewer that this is highly likely, and we have amended the text to simply state that the slowly increasing phase of NADP^+^ reduction kinetics (rather than any change in enzymatic efficiency) is on the same time scale. In addition, this possibility is presented in Figure 6—figure supplement 1.

The authors should consider the recent finding that the NADH-1 complex appears not to use NADPH, but Fd which would be in line with their interpretation.

We have now incorporated this information.

[Editors' note: further revisions were suggested prior to acceptance, as described below.]

Essential revisions:The reviewers judged that the current revisions corrected most of the issues, mainly by eliminating some areas of over-interpretation and narrowing the focus of the discuss. However, some important problems remain, especially regarding the interpretation of the NADPH florescence kinetics. As brought up by reviewer #1 in the first review, and further expanded on be reviewer #2 in the second review, the biphasic characteristics of the fluorescence rise can have multiple interpretations, and this issue has not been addressed in the revised text, which still sticks to an over-simplistic interpretation and the text needs to introduce other possible interpretations especially because the data presented thus far cannot exclude them. The reviewer and editor feel that an acknowledgement and full discussion of these possibilities is not only essential for the science, it will strengthen the paper substantially.The paper may be acceptable for publication if this, and the smaller issues brought up by reviewer #2, can be addressed. In particular, the revision must address the inaccurate interpretation of the Kauny/Setif reference and discuss possible alternative interpretations of the fast and slow phases of the NADPH kinetics.

This comparison with cyanobacteria is no longer made. We have included new paragraphs discussing other interpretations of the slow phase of NADPH kinetics in the results, discussion and Appendix. Moreover, we have removed the final paragraph of the Discussion to temper the argument.

Reviewer #2By and large the questions have been mostly addressed, however some problems remain regarding the NADPH florescence kinetics, specifically their biphasic characteristics of their rise. The authors still have not excluded other downstream processes as being responsible for the slow phase. These can be slowly changing fast rates of redox sinks. In other words, these can be relatively fast NADPH consumption process (very fast relative to the times scales of the experiment), but their rates can be slowly changing. The argument post about the lack of biphasicity cyanobacteria (Kauny/Setif reference) is not accurate: although clearly different kinetics, with different shapes are involved in cyanobacteria, the slower phases are clearly visible and actually correlate to things like state transitions and the activation of the Calvin cycle. Accordingly, processes downstream of PSET (but interacting due to back-propagation effects) seem to be responsible for these multi second kinetic features.

The discussion of cyanobacterial NADP(H) kinetics has been removed from the Appendix text.

Since there are intact chloroplasts, it is clearly possible such effects are present in the current experiments. Thus the argument based on measuring the abundance of electron transport chain components is not sufficient to strongly argue against the dominance of downstream processes in governing the slower phase. This needs to be clearly articulated in the main results text as well as in discussion. That said, the hypothesis that relocation accounts for these phases is not excluded either and altogether the interpretations seem to hold together. Nevertheless, the cited weakness needs to temper the strength of the arguments.

We agree, and on reflection this should have been spelled out more clearly in the text. We have now introduced this possibility in the Results section, and expanded on it in more detail in the discussion and Appendix 1.